# Robust Image Processing Framework for Intelligent Multi-Stage Malaria Parasite Recognition of Thick and Thin Smear Images

**DOI:** 10.3390/diagnostics13030511

**Published:** 2023-01-31

**Authors:** Thaqifah Ahmad Aris, Aimi Salihah Abdul Nasir, Wan Azani Mustafa, Mohd Yusoff Mashor, Edy Victor Haryanto, Zeehaida Mohamed

**Affiliations:** 1Faculty of Electrical Engineering and Technology, Universiti Malaysia Perlis, UniCITI Alam Campus, Sungai Chuchuh, Padang Besar 02100, Malaysia; 2Advanced Computing (AdvCOMP), Centre of Excellence, Universiti Malaysia Perlis, Pauh Putra Campus, Arau 02600, Malaysia; 3Faculty of Electronic Engineering and Technology, Universiti Malaysia Perlis, Pauh Putra Campus, Arau 02600, Malaysia; 4Faculty of Engineering and Computer Science, Universitas Potensi Utama, Medan 20241, Indonesia; 5Department of Medical Microbiology and Parasitology, School of Medical Sciences, Universiti Sains Malaysia, Kubang Kerian 16150, Malaysia

**Keywords:** plasmodium parasite, image segmentation, thresholding, clustering, species and staging, machine learning

## Abstract

Malaria is a pressing medical issue in tropical and subtropical regions. Currently, the manual microscopic examination remains the gold standard malaria diagnosis method. Nevertheless, this procedure required highly skilled lab technicians to prepare and examine the slides. Therefore, a framework encompassing image processing and machine learning is proposed due to inconsistencies in manual inspection, counting, and staging. Here, a standardized segmentation framework utilizing thresholding and clustering is developed to segment parasites’ stages of *P. falciparum* and *P. vivax* species. Moreover, a multi-stage classifier is designed for recognizing parasite species and staging in both species. Experimental results indicate the effectiveness of segmenting thick smear images based on Phansalkar thresholding garnered an accuracy of 99.86%. The employment of variance and new transferring process for the clustered members, enhanced *k*-means (EKM) clustering has successfully segmented all malaria stages with accuracy and an F1-score of 99.20% and 0.9033, respectively. In addition, the accuracies of parasite detection, species recognition, and staging obtained through a random forest (RF) accounted for 86.89%, 98.82%, and 90.78%, respectively, simultaneously. The proposed framework enables versatile malaria parasite detection and staging with an interactive result, paving the path for future improvements by utilizing the proposed framework on all others malaria species.

## 1. Introduction

In this day and age, malaria remains a threat to public health worldwide, especially in tropical regions, such as Africa. This can be due to the environmental conditions suitable for the growth of mosquitoes and exacerbated by poor socio-economic conditions. Malaria is caused by plasmodium parasites transmitted through the infected female *Anopheles* mosquito bite. There are five species of malaria, namely *P. falciparum*, *P. vivax*, *P. malariae*, *P. ovale*, and *P. knowlesi*. However, among these species, *P. falciparum* and *P. vivax* species are the two most common species that typically cause malaria infection, while others are considered to be occasionally seen. As stated in World Health Organization (WHO) report in 2020, there were an estimated 241 million malaria cases and 627,000 malaria deaths worldwide [1]. Additionally, 77% of all malaria deaths occurred in children under five years old and remained the highest recorded mortality rate in 2020 [1].

Despite advances in rapid point-of-care diagnostics, rapid diagnostic tests (RDTs), and manual microscopic examination based on thick and thin blood smears remain the most recommended diagnostic methods for routine malaria diagnosis of suspected malaria cases. On top of that, microscopic examination is the “gold standard” method for malaria diagnosis [2]. In this procedure, both thick and thin blood smears are used to identify the parasites and counted to quantify the level of parasitemia [3]. Here, the analysis of a thick blood smear is used for parasite detection. This is followed by the analysis of a thin blood smear to identify malaria parasite species and its life-cycle stages. Thick blood smears utilize a greater volume of blood and multiple layers of red blood cells (RBCs), which is about 6 μL of blood, making it more sensitive for parasite detection [3]. Meanwhile, thin blood smears utilize a single layer of uniformly spread RBCs, which is only about 2–3 μL of blood, making it unsuitable for parasite detection [3]. However, due to alcohol fixation prior to staining, the RBCs in the thin blood smear are preserved, which makes it more sensitive to analyze parasite morphology and parasitic forms that may be present. 

Currently, lab technicians will manually count and examine thick and thin blood smear samples under a microscope. Although the manual microscopic examination is acknowledged for its high sensitivity and being competently specific, the sensitivity and specificity of microscopy differ based on the quality of the stained slide and the expertise of the lab technicians. Nevertheless, this manual evaluation may vary amongst slide readers and be inaccurate [4]. Furthermore, this procedure is labor-intensive. The lab technicians usually examine 30 to 40 slides per day while being responsible for diagnosing other diseases [5]. In addition, highly skilled lab technicians are needed to prepare and examine the slides, and distinguish between the parasite species and stages. This is because some parasite morphologies are almost similar, such as those of *P. vivax*, *P. ovale* species, and the early trophozoite stage of *P. falciparum* and *P. knowlesi* species [6]. Moreover, microscopic examination and RDTs have limitations, which are ineffective when detecting sub-microscopic infections. Therefore, due to the shortcomings of the currently recommended diagnostic methods, new diagnostic procedures are compulsory.

Thus, a computerized system for malaria diagnosis consisting of image processing and machine learning approaches is necessary to examine more blood slides compared to the manual diagnosis. Yet, several issues arise upon developing a computerized system, such as the emergence of noise, shadow, random backgrounds, overlapping objects, and illumination issues, and making segmenting medical images, such as blood images, and difficult work in image analysis and processing [7]. In addition, the low image quality and the usage of staining solution significantly impact the parasite’s appearance, contributing to the false analysis in the computerized system. In addition, the small size of the parasites is also one of the main factors that make the computerized system difficult to isolate the parasites from stained blood images on thick blood smear samples. Thus, relevant image-processing procedures and machine learning algorithms are necessary to provide good results in analyzing thick and thin blood smear images. Recently, many studies have been conducted to diagnose malaria based on thick and thin blood smear images with various transfer learning techniques. Hence, novel network architectures and image-processing procedures were proposed to improve the performance of malaria diagnosis [8,9,10]. 

Azif et al. [11] reported accuracy and sensitivity of 61.53% and 98.04%, respectively, for parasite detection based on thick blood smear images. Correspondingly, Gitonga et al. [12] reported 96.2% accuracy in identifying plasmodium species and 99.9% classification of life-cycle stages on infected RBC images using an artificial neural network (ANN) classifier. On the other hand, Fatima and Farid [13] reported an accuracy of 91.80% for parasite detection based on thin blood smear images using a segmentation approach of bilateral filter and adaptive thresholding technique. Additionally, Parveen et al. [14] proposed a multilayer perceptron (MLP) neural network to classify malarial parasites into three different species. The study reported accuracy of 85% was achieved when performing the classification using the proposed classifier. Alternatively, Nugroho et al. [15] have identified three malaria stages of *P. falciparum* species by utilizing MLP training by the backpropagation algorithm. The results indicate that the proposed method achieves an accuracy of 87.8%, a sensitivity of 81.7%, and a specificity of 90.8% for detecting infected RBCs.

Certainly, the segmentation process is important since further steps, such as feature extraction and classification, depend on the accurate segmentation of the region of interest (ROI). Therefore, a good image segmentation procedure that can be used for obtaining segmented malaria parasites in thick and thin blood smear images is required to be analyzed thoroughly. Furthermore, segmentation quality is important for the success of the image analysis process. Many researchers have proposed various image segmentation approaches to improve malaria diagnosis. This includes the application of *k*-means (KM) clustering [16], fast *k*-means (FKM) clustering [17], cascaded EKM [18], fuzzy *c*-means (FCM) clustering [18], watershed distance transform [19], Otsu’s thresholding [20], and adaptive thresholding [21]. In addition to segmentation, a good image classification technique is also important in producing a reliable malaria diagnosis system. It can be used to classify malaria parasites in thick blood smear images and types of species in thin blood smear images. In addition, it can also classify stages in thin blood smear images required to diagnose malaria efficiently. 

The significance and inspiration for conducting this study have been described in this paragraph. The recent advances in machine learning have provided an opportunity to utterly explore automating the detection of parasites from thick and thin smears images. Nonetheless, the literature lacks proposed works to detect and segment malaria parasites in thick and thin smears based on one complete standard framework. Most of the work presented focused on either detecting thick parasite smears for counting or classifying parasites extracted from thin smear images based on frameworks presented in different studies. Furthermore, to the best of the authors’ knowledge, most of the existing references have paid significant attention to processing only the thin blood smear samples compared to the thick blood smear [22,23,24]. As for the thick smear, most of the literature has focused on counting the parasites to estimate the level of parasitemia after the segmentation is complete, lacking a framework for enhancing the parasite count with machine learning. Despite significant improvements in model performance in thin smear analysis, there is currently no widely used tool for the automated detection and staging of malaria parasites.

The main objective of this research work is to develop an intelligent multi-stage malaria parasite recognition for microscopy diagnosis of both thick and thin smear images. Additionally, the sub-objective of this research is to find the best image segmentation technique to segment thick and thin smear images, find the best classifier to classify malaria parasites in both thick and thin images, and classify species and stages in thin smear images. Note that this research will focus on processing the most common species responsible for malaria infection in most cases. They are *P. falciparum* and *P. vivax* species of thick and thin smear samples of the ring, trophozoite, schizont, and gametocyte stages. In the experimental analysis, eleven relevant image segmentation techniques and three machine learnings have been compared to select the most suitable approach for malaria parasite recognition with the purpose to improve the standardization on thick and thin smear inspection for malaria research and potential field diagnosis. The contributions of this paper can be explicitly stated below:A full framework constituted from image processing and machine learning is designed to address the problem of detecting parasites in thick smear microscopy, classifying parasite species, and staging for thin smear microscopy.A standard segmentation framework based on thresholding and clustering algorithms is developed to segment malaria parasites in thick and thin smears of all stages for both *P. falciparum* and *P. vivax* species.The proposed segmentation framework is applicable in producing good segmentation results, though the captured malaria image varies in terms of lighting conditions, and each stage of the malaria parasite consists of different shapes and sizes.The use of several machine-learning algorithms for classifying parasite species and life-cycle stages resulted in benchmark results that can be compared with deep-learning algorithms.

## 2. Related Literature of Malaria Diagnosis

Various applications of image segmentation procedures and artificial intelligent classifiers have been proposed by previous researchers to improve the network performances to perform malaria diagnosis, including the classification of malaria parasites, their species, and life-cycle stages. For instance, Nanoti et al. [25] have proposed malaria detection of species and life-cycle stages based on thin blood smear images. In this study, several pre-processes were applied to the input image: geometric mean filter, median filter, and forward discrete curvelet transform, to improve the image quality. Subsequently, partial contrast stretching was utilized to enhance the contrast of the parasite region. Here, the enhanced image was converted into RGB (red, green, blue), HIS (hue, intensity, saturation), and L*A*B color spaces and applied with KM clustering. Based on this study, the A*B component produces a good, segmented image. Thus, it is selected to be extracted by various texture and shape-based features, such as Gray Level Co-occurrence Matrix (GLCM), Gray Level Run Length Matrix (GLRLM), perimeter, area, entropy, local binary pattern, Hu’s seven invariant moments, and histogram features. A total of 300 images were extracted and d on the *k*-nearest neighbor and support vector machine (SVM). The finding shows that classification using *k*-nearest neighbor produces higher accuracy and sensitivity with values of 90.17% and 90.23%, respectively, compared to SVM.

Hendrawan et al. [26] have proposed image-processing procedures to identify malaria infection in four types of plasmodium species and three different life-cycle stages. First, the input image was enhanced using a color normalization process followed by gamma correction to improve the quality of the image. Consequently, edge enhancement was applied to sharpen the parasite’s edge. The image was then converted to grayscale and green color components before segmenting using fuzzy c-means (FCM) clustering. After that, erosion and closing morphological operation was conducted to remove unwanted pixels, and close the holes produced inside the parasite’s region. This proposed analysis was tested on 574 malaria image datasets. The experimental results present that the proposed image-processing procedures successfully obtain a good malaria-segmented image. It was done by applying a green color component with a segmentation performance sensitivity of 97.91%, specificity of 98.61%, and accuracy of 98.26%.

On the other hand, Dave [27] has analyzed several image-processing steps for malaria parasite detection based on thick blood smear images. At first, the malaria image was converted into hue, saturation, and value (HSV) color space, and the saturation component was extracted. Correspondingly, the average filter was applied to the saturation component to reduce the noise produced in the malaria image. After that, the image was segmented using adaptive thresholding, and several morphological operations, such as filling holes and closing operations, were applied to eliminate unwanted pixels. Using the final segmented images, various features based on shape, texture, color, and frequency domain are extracted to be fed as input to the SVM classifier. The finding shows that classification using SVM has produced a good discrepancy in parasite count, sensitivity, and specificity with values of 7.18%, 86.34%, and 96.60%, respectively.

Aggarwal et al. [28] have proposed image-processing procedures for detecting malaria parasites using thin blood smear samples. First, the input image was enhanced using the histogram equalization technique to improve the image’s contrast. Subsequently, a median filter was applied to discard noises produced on the enhanced image. The image is then converted to binary using Otsu thresholding. This proposed analysis was tested on 60 malaria image datasets. The average evaluation results demonstrate that the proposed image-processing procedures successfully obtained a good malaria-segmented image with an accuracy of 93%.

Setianingrum et al. [29] have proposed image-processing procedures to identify malaria parasites in trophozoite and schizont stages based on thin blood smear images of *P. falciparum* species. At first, the malaria image was converted into grayscale and enhanced by contrast stretching. Next, the enhanced image was segmented using Otsu thresholding. After that, the GLCM and morphological features were extracted from the malaria image to be fed as input to the SVM classifier. The result shows that SVM classification has achieved 91.67% accuracy in detecting the two stages of *P. falciparum* species.

Mustafa et al. [30] have analyzed several thresholding methods, such as Wolf, Bradley, Bernsen, Triangle, Deghost, and FCM algorithms, on the malaria image dataset to simplify the malaria image into something easier to examine. At first, the malaria image was converted to a grayscale image before applying the selected thresholding methods. This proposed analysis has been tested on 30 malaria image datasets. Note that the average evaluation results present that the FCM algorithm method successfully obtains a good malaria-segmented image by having a better segmentation performance compared to other methods with a sensitivity of 95.51% and specificity of 75.1%, and overall performance of 85.31%.

Alternatively, Nugroho et al. [31] proposed a segmentation procedure to segment infected red blood cells (RBCs) in thin blood smear images using threshold and morphological operation. In this study, the malaria image was enhanced and converted to grayscale before being applied to a multilevel Otsu threshold. Overall, the proposed procedure was tested on a total of 30 malaria. The results showed that the proposed segmentation procedure achieved 96.74% of accuracy, 76.77% of sensitivity, 99.74% of specificity, 97.84% of prediction value positive, and 96.61% of prediction value negative. They indicate that this proposed procedure provides a consistent result for segmenting parasites in thin blood smear images.

Sifat and Islam [32] have proposed identifying the malaria parasite species and life-cycle stages based on thin blood smear images. The combination of median and geometric mean filters was utilized in the proposed methodology to improve image quality. After that, the image was enhanced by applying a partial contrast technique before extracting the RBC from blood smear images using the U-Net segmentation technique. Subsequently, a convolutional neural network (CNN) was used to identify infected RBCs. In contrast, the visual geometry group 16 (VGG16) network was utilized to distinguish between the various species and the malaria life cycle. This proposed method has been practiced on 5512 malaria images. Using the CNN model, the detection accuracy and specificity of infected RBC were 100% and 95%, respectively. Meanwhile, an average accuracy of 95.55%, and specificity of 94.75% was achieved for species identification, while an overall accuracy of 96.25%, and specificity of 94.82% were achieved in the ring life cycle.

Nugroho et al. [33] proposed the classification of malaria species and the life cycle stages of *P. falciparum*, *P. vivax*, *P. malariae*, and *P. ovale* species. In this research, the input image was converted to a grayscale image. After that, the Otsu thresholding was applied to the grayscale image to produce a binary image. A morphological operation called the opening operation was applied to the binary image to remove unwanted regions. Consequently, a morpho-geometrical feature has been extracted and fed as input to Naïve Bayes (NB) classifier. The application of the proposed algorithm achieved a sensitivity value of 84.37%, a positive predictive value of 80.60%, and an F1 score of 80.60%, which shows that the proposed algorithm is reliable.

Swastika et al. [34] have employed regularization on several CNN architectures. It was to investigate the impact of regularization on increasing malaria parasite detection accuracy using thin blood smear images. This study has applied regularization methods, including data augmentation, dropout layer, and L2 regularization, to ResNet-50, MicroVGGNet, and BaselineNet-1 models. The analysis was conducted on a total of 27,588 images, consisting of 13,779 uninfected blood cells and 13,779 parasite-infected images. It was then compared between the CNN model with regularization and without regularization. The findings indicate that 94.92% accuracy was attained when BaselineNet1 was used without regularization. Meanwhile, the accuracy of 97.12%, 95.64%, and 96.28% was attained by applying regularization on ResNet-50, MicroVGGNet, and BaselineNet-1, respectively.

On the other hand, Taha and Liza [35] have utilized two types of deep learning models to identify malaria from blood smear images based on CNN. In this study, 27,558 thin blood smear images have been resized into the same size array with a width and height of 299 × 299 pixels. Two deep learning models were used and compared: basic CNN and InceptionResNetV2 model. Based on the findings, the InceptionResNetV2 provides much higher accuracy with a value of 97.10%. This includes high precision, recall, F1-score, and Matthew’s correlation coefficient, with 96.93%, 97.28%, 97.10%, and 94.20%, respectively. Compared to the basic CNN with five convolutional layers, it obtained an accuracy of 96.46% with precision, recall, F1-score, and Matthew’s correlation coefficient being 98.30%, 94.56%, 96.39%, and 92.99%. 

Widiawati et al. [36] have proposed a malaria detection method based on thick blood smear images. This study applied several pre-processing techniques, such as color model conversion, contrast stretching, and median filter, on 38 thick blood smear images. Correspondingly, the images were segmented using Bottom-Hat and Adaptive Entropy Thresholding. After that, several texture features were extracted from the green component image: mean, standard deviation, skewness, energy, and entropy. Here, three different classifiers have been compared: NB, SVM, and MLP. The best results in the classification of species were obtained from the NB classification with an accuracy of 82.05%.

Maqsood et al. [37] have analyzed the efficiency of several current deep-learning models on malaria detection using blood smear images. Additionally, the author also proposed an efficient deep-learning method for classifying infected and uninfected malaria cells utilizing CNN without using a hand-crafted feature. This study employed image augmentation and bilateral filtering techniques to emphasize the features of RBCs before training the datasets. Here, the modified CNN model is generalized, and over-fitting can be prevented because of image augmentation approaches. The analysis has been conducted on a total of 27,588 malaria datasets obtained from the National Institute of Health (NIH) website. The results indicate that the proposed algorithm can perform malaria detection with an accuracy of 96.82%.

Other than that, Setiawan et al. [38] have proposed a computer-aided diagnosis to detect and count the malaria parasites in thick blood smears. This study utilizes a color-based segmentation to detect the parasite using three different color spaces: RGB, YCbCr, and LAB. Initially, the malaria image was converted to several color components. Next, Otsu thresholding was used to segment the converted malaria image. This proposed study has been applied to 27,558 images of 13,779 infected malaria images and 13,779 uninfected malaria images. The best results are achieved by color-based segmentation in RGB color space with an accuracy of 94.75%, sensitivity of 96.02%, and specificity of 93.47%.

Rameen et al. [39] proposed supervised machine learning techniques for malaria identification based on samples of blood smears. This study involves several pre-processing, such as resizing the input images and converting the images into grayscale. Subsequently, Otsu thresholding was implemented to produce the binary image. Afterwards, the GoogLeNet was applied to extract features from the segmented blood smear image to be fed as input to the SVM classifier. Based on the experimental result, the application of the proposed algorithm achieved an accuracy value of 95.80%, which proves that the proposed algorithm is reliable.

Aris et al. [40] have analyzed the utilization of several clustering techniques to acquire fully segmented malaria parasites based on thick blood smear images of *P. falciparum* and *P. vivax* species. At first, the author applied modified global contrast stretching, followed by gray world techniques, to improve the image quality. Next, the enhanced image was converted into a blue component of RGB color spaces to ease the segmentation process. Here, KM, FKM, and EKM clustering techniques were applied to 100 thick blood smear images. Eventually, the segmentation performances were measured and demonstrated that segmentation using the FKM clustering algorithm produced a good segmentation result, with an accuracy of 99.91%, sensitivity of 75.75%, and specificity of 99.93%.

Apart from that, Shal and Gupta [41] examined the application of different classifier models to classify uninfected and infected cells for malaria detection. In this study, classifier models, namely CNN, YOLOv4, YOLOv5, and single shot detector, were applied on 27,560 images consisting of 13,780 infected images and 13,780 uninfected cell images. Furthermore, these four classifier models were tested on several performance evaluation techniques, such as accuracy, true positivity rate, and precision. Based on this analysis, the best results are achieved in YOLOv5 with an accuracy of 94.67%, a true positivity rate of 50.30%, and a precision of 95.25%. 

Razin et al. [42] developed a model for malaria parasite detection by implementing the CNN and YOLOv5 model to detect the malaria parasites and classify the uninfected and infected malaria images. This research was applied to a total of 27,588 thin blood smear images obtained from the NIH website, a publicly accessible dataset. In accordance with the experimental outcomes, the YOLOv5 model obtained an accuracy of 95.37% for infected and 97.05% for uninfected samples. In addition, the CNN model can obtain the highest accuracy with a value of 96.21% for both infected and uninfected blood samples. Additionally, this CNN model also achieved high precision and recall rates of 95.42% and 97.05%, respectively. 

On the other hand, Zarima et al. [43] utilize hand-crafted features and deep learning techniques on blood smear images to diagnose malaria. In this study, a texture feature known as Local Binary Pattern (LBP) has been extracted from a total of 27,588 images, consisting of 13,779 uninfected blood cells images and 13,779 parasite’s infected images. This study analyzed several deep-learning models, namely ResNet-34, VGG16, Inception V3, and EfficientNet, to determine which model yield the best accuracy. Based on the experimental result, the EfficientNet model has obtained the highest accuracy, with a value of 91%, compared to the VGG16, ResNet-34, and InceptionV3 models, with 87%, 81%, and 77%, respectively. 

Gummadi, Ghoosh, and Vootla [44] employed a transfer-learning-based CNN architecture to differentiate between uninfected and infected malaria images using global average pooling (GAP) and heat maps. In this study, a total of 27,588 thin blood smear images have been used on three transfer learning-based neural network architectures, known as VGG16, VGG19, and InceptionResNetV2 architecture. These three classifiers were trained and compared. Based on the study finding, InceptionResNetV2 architecture yielded a maximum accuracy of 96.88%, with sensitivity, specificity, F1-score, and precision of 97.35%, 96.41%, 96.89%, and 96.44%, respectively. In addition, this study also interfaced with a newly developed web service that anyone can easily access.

Abdurahman, Fante, and Aliy [45] have proposed an automated malaria parasite detection based on thick blood smear images using modified YOLOv3 and YOLOv4 models. In this study, these modified models were called YOLOv3-MOD1, YOLOv3-MOD2, and YOLOv4-MOD. The effectiveness of the modified YOLOv3 and YOLOv4 models was assessed using a total of 1182 blood smear images consisting of 948 malaria-infected images and 234 normal images. Based on the study verdict, YOLOv3-MOD1, YOLOv3-MOD2, and YOLOv4-MOD models have superior mean average precision (mAP), recall, precision, F1-score, and average IOU compared to their original versions, faster region-based convolutional neural network (FR-CNN), and single shot multi-box detector (SSD). However, among all the models, YOLOv4-MOD has the highest detection accuracy, with a mAP of 96.32%, recall of 94%, precision of 95%, F1-score of 94%, and average IOU of 62.12% as compared to YOLOv3-MOD1 and YOLOv3-MOD2 models.

Based on the findings from previous researchers’ work, various segmentation techniques have been proposed and successfully applied for the segmentation of malaria parasites or infected RBCs. In addition, it can be found that segmentation using thresholding and clustering offers several advantages in segmenting malaria images. Generally, the thresholding algorithm is suitable to be applied when the objects are not overlapped, and the gray levels are distinct from the background level. However, this algorithm is less suitable to be applied on images blurred at the object boundaries or for the overlapped ROI area as it will result in over-segmentation. Meanwhile, the clustering algorithm will segment the image into clusters, having pixels with similar characteristics, such as grouping the pixels based on intensity similarity. As a result, both thresholding and pixel-level segmentation based on clustering will also be added to the framework for segmenting the malaria parasites. This is an initiative to find suitable segmentation techniques, reducing the confounding of neighboring cells on parasite detection due to over-segmentation.

Subsequently, based on the review of intelligent classifiers, it is found that several research has utilized SVM to perform the malaria detection process and classify malaria parasite species and their life-cycle stages. Although SVM is a popular method utilized to classify malaria parasites due to its good generalization performance, SVM has some limitations. The biggest limitation of SVM lies in the selection of the kernel [46]. Therefore, SVM will not be utilized for classification in this study due to these limitations. Additionally, the performance of the extreme learning machine (ELM) network will be compared with other types of intelligent classifiers, online sequential-ELM (OS-ELM), and RFs, to determine the capability of these methods for the classification process. The OS-ELM has been chosen to be compared because it is an upgraded version of ELM. It has been proven to be faster than the other well-known sequential algorithms and produces better generalization performance with lower training time [47]. As for RF, it is an ensemble classifier that can be used for complex classification tasks.

## 3. Materials and Methods

In this study, the processes to obtain thick and thin blood smear segmented images and classify malaria parasites, types of species, and their life-cycle stages consist of three main phases. The first phase deals with data collection and image acquisition setup, while the second deals with segmentation. Lastly, the third phase deals with classification. This section describes various applications of image segmentation and machine learning techniques applied to thick and thin blood smear images. The full proposed procedures for thick and thin blood smear images included image enhancement, extraction of color components, image segmentation, watershed segmentation (only for thick blood smear images), removal of unwanted pixels, feature extraction, and classification will be explained in next sections.

### 3.1. Data Collection and Image Acquisition Setup

In this research, the malaria slides of *P. falciparum* and *P. vivax* species of both thick and thin blood smear samples were obtained from the Department of Medical Microbiology and Parasitology, Hospital Universiti Sains Malaysia (HUSM). The blood slides were examined under 100× oil immersion using an *OLYMPUS BX41* digital microscope to capture the thick blood smear images, as shown in Figure 1. The images of the slides were captured using an *OLYMPUS XC50* digital camera with a resolution of 1294 × 980 pixels and saved in a jpg format. The blood slides were examined under the 100× oil immersion objective of the *Leica DLMA 1200* digital microscope for the thin blood smear images. The images of the slides were captured using a *Luminera Infinity-2* digital camera with a resolution of 800 × 600 pixels and saved in bitmap (*.bmp) format. 

This study captured 500 thick blood smear images from four malaria slides containing both *P. falciparum* and *P. vivax* species. Meanwhile, 4035 thin blood smear images with various conditions were captured from 34 malaria slides consisting of both *P. falciparum* and *P. vivax* species in the ring, mature trophozoite, schizont, and gametocyte stages. Additionally, both thick and thin blood smear images were captured under several lighting conditions, such as underexposed, overexposed, and normal, as seen in Figure 2. Figure 2 represents the samples of the captured blood images from *P. falciparum* and *P. vivax* in thick and thin blood smear samples. As seen in Figure 2, the stain highlights the malaria parasites and other components inside the blood sample, such as white blood cells (WBCs), platelets, and artifacts. As a result, the WBCs and platelets also share a similar color property with the malaria parasites, as seen in Figure 2b,f. Figure 2d,e,i exhibit the samples of the captured malaria images with the presence of Herring-bone scratches in the glass slide, air bubbles under the coverslip of the slide, and artifacts, respectively. 

### 3.2. Image Segmentation Using Various Local Thresholding and Clustering Techniques on Thick and Thin Blood Smear Images

Image segmentation separates an image into homogenous segments based on specific criteria [48]. The basic aim of this separation was to ensure the images were easily analyzed and interpreted by preserving the image quality. Note that segmentation quality is one of the important elements for the success of the image analysis process. Segmentation, such as thresholding and clustering, offers several benefits compared to other methods. The significant benefit of the thresholding technique was that it could be employed on non-overlapping objects because their grey levels were easily distinguished from the background level. Meanwhile, the significant benefit of the clustering technique was ideal for segmenting the medical image as the number of clusters could be determined from the image with specific anatomical regions. Therefore, this research used several local thresholding and clustering techniques to segment malaria images. This local thresholding technique was superior in segmenting uneven lighting images because this thresholding technique determines the threshold values according to the local area information. Explanations about the types of thresholding and clustering algorithms were presented as follows:

**Adaptive thresholding**: Local adaptive thresholding was a basic and simple algorithm to separate the foreground from the background with non-uniform illumination. For each pixel in the image, a threshold had to be calculated. If the pixel value is below the threshold, it is set to the background value. Otherwise, it assumes the foreground value. The assumption behind this method is that the smaller image regions were more likely to have approximately uniform illumination, being more suitable for thresholding. Here, the input image was partitioned into several intersecting sub-images. After that, the histogram of each sub-image was analyzed to discover the ideal threshold for that particular sub-image. By approximating the outputs of the sub-images, the threshold for each pixel was obtained. The neighborhood size had to be large enough to cover good foreground and background pixels; otherwise, a poor threshold was chosen. On the other hand, choosing regions that are too large could violate the assumption of approximately uniform illumination. In the adaptive thresholding method, the local threshold was determined as the mean of the local intensity distribution of each pixel [49]. The threshold for each pixel was computed by:(1)$$T=\frac{max+min}{2}.\;$$

**Niblack**: Niblack algorithm was one of the most popular local adaptive algorithms. It determines the threshold value of every pixel by sliding a particular window size throughout the image pixel location to set the threshold value. It is performed according to the local standard deviation and local mean [50]. The threshold was obtained based on the following equation:(2)T=m+kσ.

Here, *k* is referred to as a constant value of −0.2 fixed by the Niblack algorithm author, while the window size could be determined manually. Therefore, both sliding window size and the *k*-factor parameter might affect binarization quality. However, this method had a drawback: It makes excessive noises in the emptied windows [51]. Additionally, the Niblack algorithm did not suffice to adapt non-uniform illumination images. 

**Sauvola**: Sauvola algorithm was inspired by the Niblack algorithm, which works by computing the threshold value using a dynamic range of standard deviation in the gray value image [52]. By implementing various gray-level values in the images, this method addresses the Niblack weakness of excessive noises based on the standard deviation effect. In Sauvola, the interpolation was utilized for the remaining pixels after the initial threshold had been determined. This approach gives a more precise threshold value for each pixel. Thus, using the following equation, the threshold value was determined:(3)$$T=m\left(1-k\left(1-\frac{\sigma }{R}\right)\right).$$

Here, *k* is referred to as a factor that took on positive values and was often set to 0.5, while *R* was the dynamics of the standard deviation with a fixed value of 128. In addition, the noise issue was resolved by this algorithm, but it was unable to perform well on the low-contrast image. In the Sauvola method, the threshold was computed similarly to the Niblack method. Still, it used the average of an image’s highest and lowest gray value to modify the local standard deviation to emphasize segmentation on low-contrast information images. 

**Bradley**: Bradley algorithm utilized adaptive thresholding to adjust the spatial differences in illumination by employing the integral image of the input data [53]. This method is effective and easy to use and is more vulnerable to variations in image illumination. To produce the integral image, I(x,y) should be maintained at each position as the sum of those f(x,y) terms toward the left being above the pixel (x,y), which can be computed by the following equation for each pixel:(4)$$\sum_{x=x_{1}}^{x_{2}}\sum_{y=y_{1}}^{y_{2}}f\left(x,y\right)=I\left(x_{2},y_{2}\right)-I\left(x_{2},y_{1}-1\right)I\left(x_{1}-1,y_{2}\right)+I\left(x_{1}-1,y_{1}-1\right).$$

This algorithm computes the average window size of pixels centered on each pixel. The average calculation was completed in constant time by utilizing an integral image. This approach computed the integral image using the input image in the first phase. In a subsequent phase, a comparison analysis was used to determine whether a pixel belonged to the foreground or background region. This was done by determining the window size average for each pixel using the integral image in constant time. Moreover, inconsistent luminance images could be effectively binarized using this algorithm. Finally, the Bradley method determines the local threshold based on the mean of the neighboring pixels using the integral image as the input. 

**Phansalkar**: Phansalkar algorithm is the modification of Sauvola’s method, which was invented to overcome Sauvola’s weakness, specifically to deal with low-contrast images [54]. In Sauvola’s method, if the contrast in the local neighborhood were low, then the local mean values would be lower, and the threshold value would fall below the mean, causing the dark regions of the background to be removed. The Phansalkar algorithm functions similarly to Sauvola’s method, which is based on the local mean and employs the combination of the local standard deviation and parameters to constrain the calculation in certain conditions. In addition, this method worked similarly to Sauvola’s method for high values of the local mean. However, in Phansalkar, if the value of the local means is low, then the threshold value determined should be much higher than the value of the local mean and the value determined by Sauvola’s method. This method could be done by computing the threshold value using the following equation:(5)$$T\left(x,y\right)=m\left(x,y\right)\left[1+pe^{-q.m\left(x,y\right)}+k\left(\frac{s\left(x,y\right)}{R}-1\right)\right],$$
where k, p, and q were constants. Here, the value of q was chosen above the value of the local mean. On the other hand, the exponential term in the equation becomes negligible as the equation functions similarly to Sauvola’s method. Subsequently, the constant p decides the magnitude to which the exponential term affects the threshold. If the p values were low, such as (0<=p<=1), then it would function similarly to Sauvola’s method. However, if the p value is high, such as (p>5), the threshold would become high, and too many background pixels would be classified as foreground. 

**Feng**: Feng’s method was built on the idea of two local windows [55]. This algorithm functioned locally and calculated the dynamic range of gray-level value standard deviation. Here, the values of the minimum gray level, standard deviation, and local mean were calculated in the primary local window to determine the local threshold value. Meanwhile, to compensate for the negative effect of uneven illumination, the dynamic range standard deviation was calculated in the secondary window as it could tolerate different degrees of uneven illumination. Therefore, the respective equation was used to calculate the threshold value:(6)$$T=\left(1-a_{1}\right)m+a_{2}\left(\frac{s}{R_{s}}\right)\left(m-M\right)+a_{3}M,$$
where $a_{1}$ is between 0.1–0.2, while $a_{2}=k_{1}{\left(\frac{s}{R_{s}}\right)}^{\gamma }$ and $a_{3}=k_{2}{\left(\frac{s}{R_{s}}\right)}^{\gamma }$. These $a_{1},\;a_{2}$, and $a_{3}$ were defined as adaptive weighting. The values of other parameters, such as $k_{1}$ and $k_{2}$ were proposed in the ranges between 0.15–0.25 and 0.01–0.05, respectively. In the Feng algorithm, the local threshold value was bilinearly interpolated to obtain the threshold values for all pixels to minimize the computational load. In Feng’s method, the local threshold was determined by considering two local windows, such as primary and secondary. It calculates the standard deviation of the gray value from the whole image using both windows. 

**Nick**: Nick’s algorithm was a modified version of the Niblack and Sauvola algorithm [51]. This algorithm was created to solve the limitation in the Niblack algorithm regarding excessive noises in the emptied windows and the problem in the Sauvola algorithm, which is unable to perform well on the low-contrast image. The thresholding formula of this algorithm was derived based on the basic Niblack algorithm. Apart from that, this algorithm had been improvised from the Niblack and Sauvola algorithm by shifting the thresholding value downward [51]. The following equation was used to determine the thresholding value:(7)$$T=m+k\sqrt{\frac{\sum^{\;}pi^{2}-m^{2}}{N}}.$$

Here, *k* refers to a constant parameter with a value between −0.1 to −0.2, pi refers to the value of the grayscale pixel levels in the image, and *N* is the total number of pixels in the image. The value of the *k* factor could vary from −0.1 to −0.2 depending on the application requirement. If the *k* value is close to −0.2, the noises were almost eliminated but could rupture the ROI. Meanwhile, with values close to −0.1, some unwanted pixels might remain still, but the ROI could be extracted unscathed. Additionally, the need to determine the *k* factor manually is also considered as this algorithm’s drawback. In the Nick method, the local threshold was determined based on the mean and square root of the difference between the sums of the squares of pixel intensities inside a local window. 

***K*-means**: KM clustering algorithm was one of the most popular unsupervised and iterative learning algorithms because of its simplicity. Throughout the process of employing KM clustering for image segmentation, each pixel in the image was allocated to the closest cluster center. Note that the distance between the pixel and the center was computed based on the Euclidean distance [56]. Due to its simplicity, KM clustering has become popular for many clustering tasks. However, the KM clustering algorithm had some limitations that normally arise, such as the possibility of converging to local minima, dead centers problem, and being sensitive to initialization of cluster centers. 

**Fuzzy *c*-means**: FCM clustering was conducted by designating a membership function towards each data point according to every cluster center on the justification of distance measures between the data point and cluster center. Instead of allocating each data to a single center as in the KM algorithm, each data in the FCM clustering algorithm were given a membership grade representing the amount it belongs to each center [57]. However, due to the fuzzification in the FCM algorithm, certain data points could fall to more than one cluster. Yet, FCM performs effectively for overlapping data sets in comparison to KM. 

**Fast *k*-means**: FKM clustering was developed to get through the drawbacks of the KM clustering algorithm. According to the traditional KM clustering algorithm, the distance value in each loop should be determined between each pixel and each clustering core [58]. Hence, a lot of time was consumed by this event. Therefore, the FKM clustering algorithm used the histogram value and the KM algorithm’s discrete function to reduce time consumption for training image cluster centers, and also got control of the retraining cluster center problem. The exact explanation regarding FKM clustering steps can be referred to in [57].

**Enhanced *k*-Means**: The EKM clustering was a modified clustering algorithm that was enhanced by the conventional KM clustering algorithm. This clustering technique was a part of the author’s previous work proposed by Nasir et al. and published in [59]. Here, EKM clustering has been invented to address the shortcomings of the KM clustering algorithm regarding the assignation of the cluster center. Furthermore, to facilitate the distribution of data to the relevant cluster to generate the finest segmented image, clustering has invented a new transferring mechanism for the clustered members that employ the idea of variance [59]. In addition, the variance value was calculated and often checked through the clustering process to assess the relationship between the center and its members. This action should be done because the data assigned to the center that did not meet a predetermined criterion should naturally be relocated to the appropriate cluster. A detailed explanation of how the EKM clustering algorithm works can be referred to in [59].

### 3.3. Proposed Segmentation Procedures for Thick and Thin Blood Smear Images

This section provides detailed information regarding the proposed image-processing procedures applied to thick and thin blood smear images. Thus, the full framework based on image-processing techniques that were designed to segment both thick and thin blood smear images was disclosed in the following sub-sections. 

#### 3.3.1. Thick Blood Smear Segmentation Procedure

The framework of the proposed image-processing procedure for thick smear segmentation is shown in Figure 3. In this study, the thick smear images were captured under several lighting conditions, such as underexposed, overexposed, and normal, which may result in low-quality images. To obtain a fully segmented malaria parasite region, several image-processing techniques were utilized to process the thick smear images. These processes include image enhancement, color conversion to extract color components, and image segmentation.

Here, the image enhancement technique was applied to malaria images for two main purposes. First, it was to improve the visual quality of malaria images, such as blurriness or poor contrast. Second, it was to reduce the effect of staining variation on the malaria image. Therefore, the image enhancement process was divided into two steps, namely contrast enhancement and color constancy. Contrast enhancement was performed on malaria images to enhance the region contrast of malaria parasite for easing the segmentation process.

In contrast, color constancy was performed on malaria images to standardize the color of background areas to appear as if they were captured under similar color staining. In the image enhancement process, the thick smear images were first enhanced using the modified global contrast stretching (MGCS) technique to increase the image contrast [59]. MGCS is the modified version of global contrast stretching (GCS) techniques. Additionally, MGCS techniques work based on the computation of total pixels in a histogram. The selected minimum and maximum percentage pixel values within the RGB image’s histogram were used to enhance the ROI. Note that the required parameter of minimum and maximum percentage values of *minp* = 0.1 and *maxp* = 10 was selected to produce new minimal and maximal RGB values for the contrast stretching process. The MGCS algorithm was described in more depth in [60]. After that, the color constancy technique, known as the gray world, was applied to the MGCS image. It was to remove color casts and portray the true colors of the malaria parasites by standardizing the color of background areas in malaria images to appear as if they were captured under similar color staining. The gray world was operated based on the presumption that the average pixel values for R, G, and B components will converge to the same mean value [61]. This presumption allows the elimination of color casts and the image’s color to be uniformly comparable to the actual scene.

Once the malaria image was enhanced, the image proceeded with color conversion to extract the color components for easing the segmentation process. Based on the initial analysis and the findings presented in [17], the blue (B) color component was extracted from the RGB color model to facilitate the segmentation process. Afterwards, the B components were segmented by 11 segmentation techniques consisting of local thresholding and clustering techniques. Adaptive thresholding, Bradley, Niblack, Nick, Sauvola, Phansalkar, Feng, KM, FCM, FKM, and EKM clustering were the segmentation algorithms used on the thick smear images, in which the performance of local thresholding methods depends on the window size selection. Windows sizes were used to get the local information by subdividing the images into several blocks. In the adaptive thresholding method, the selected parameter of window size was 65 × 65. As for Bradley, Nick, Sauvola, and Phansalkar, the selected window size parameter was 15 × 15. In the Niblack method, the selected parameter of window size was 250 × 250. Meanwhile, in Feng’s method, the local threshold was determined by considering two local windows, such as primary and secondary. The selected parameter of the primary window size was 5 × 5, and the secondary window size was 150 × 150.

Meanwhile, the performance of clustering methods depended on the selection of the cluster number. The clusters for KM, FCM, FKM, and EKM clustering were set into three clusters, representing the malaria parasites, artifacts, and background region. Thus, watershed segmentation with the Euclidean distance transform approach was used after the segmentation process to separate the touching malaria parasite cells. The Euclidean distance transform calculates the inequality within the neighborhood pixel to separate each touching parasite.

After isolating the touching malaria parasites using watershed segmentation, some unwanted pixels, such as artifacts, staining precipitation, and WBCs, were still encountered in the segmented images. These unwanted pixels were removed to enhance the segmentation performance and avoid misidentifying the parasites in the further classification process. In order to proceed with this procedure, two operations were applied to the resultant segmented images. The first operation was to obtain the total pixel’s area in the ROI, where the ROI referred to the malaria parasite. Here, the parasite’s size was one of the crucial morphological characteristics used to differentiate between the malaria parasite and non-malaria parasite. Meanwhile, the second operation was to remove the unwanted pixels after the segmentation process. In order to obtain the total pixel’s area in the ROI, a regional feature descriptor was used to extract the area in pixels of the segmented malaria parasite. Based on this operation, it was found that a typical malaria parasite size might have a value of the area in pixels between 7 to 280 pixels. Thus, any regions that were not in these pixel ranges were labeled as non-malaria parasites. By utilizing the morphological operation process, any pixels lower than 7 pixels and higher than 280 pixels were removed from the segmented image to obtain the final segmented image.

#### 3.3.2. Thin Blood Smear Segmentation Procedure

The framework of the proposed image-processing procedure for thin smear segmentation is shown in Figure 4. In this study, the thin smear images were captured under various conditions, as mentioned in Section 3.1. To obtain a fully segmented malaria parasite region from thin smear samples, several image-processing techniques were utilized to process the thin smear images. These processes include image enhancement, extraction of color components, image segmentation, and removal of unwanted pixels.

An image enhancement technique was applied to thin smear images to improve the visual quality of thin smear images. In addition, the thin smear images were also enhanced for the image enhancement process using the MGCS technique to increase the image contrast. Note that the required parameter of minimum and maximum percentage values of *minp* = 0.5 and *maxp* = 10 were selected to produce new minimal and maximal RGB values for the contrast stretching process. After the malaria image was enhanced, color conversion was used to extract the color components and facilitate better segmentation. As for the thin smear image, the green (G) component information was extracted from the enhanced image. Subsequently, the G components were segmented by 11 segmentation techniques consisting of local thresholding and clustering techniques. Here, the segmentation algorithms used on the thin smear images were adaptive thresholding, Bradley, Niblack, Nick, Sauvola, Phansalkar, Feng, KM, FCM, FKM, and EKM clustering. As mentioned previously, the performance of local thresholding methods depends on the window size selection. In the thin smear image, the window size selection for adaptive thresholding, Bradley, Nick, Sauvola, Phansalkar, and Niblack, including the Feng method, were 175 × 175. Meanwhile, the cluster number for KM, FCM, FKM, and EKM clustering was set to the value of three clusters, which stand for the malaria parasites, artifacts, and background region. 

After the segmented image was obtained through the image segmentation process, some unwanted regions, such as small background pixels, were still encountered in the image. Therefore, the median filter was used as a noise removal to remove these small background pixels from the segmented image. It was performed by replacing a current pixel’s value with its neighboring pixels’ median value. In addition, the median filter could also be used to fill the small holes that might appear on the segmented malaria parasite due to its good smoothing performance and preserving the edge well without blurring it. Here, the neighborhood of *n*
×
*n* (*n* = 7) pixels was used because large neighborhoods produce more severe smoothing. After segmentation and filtering processes were performed on the malaria image, there was a tendency for the unwanted regions, such as segmented RBCs, platelets, and artifacts, to appear on the segmented image. This can be due to some properties, such as color and size, which these unwanted regions share with the malaria parasite. Thus, these unwanted regions should be excluded from the segmented image to enhance the segmentation performance and to avoid the misdiagnosis of the infected cell in the further classification process. 

In order to reduce this problem, a modified version of the conventional seed-based region growing algorithm, namely the seeded region growing area extraction (SRGAE) algorithm proposed by Harun et al. [62], was applied to the segmented image. This algorithm was applied for two main purposes. First, calculate the total area in pixels for the ROI, where the ROI is referred to as the malaria parasite. Second, remove any unwanted regions that were bigger in size that cannot be cleaned using the 7 × 7 pixels median filter. Based on analysis of malaria parasites that had been segmented manually in several malaria images, it had been found that a typical malaria parasite (ring and mature trophozoite of *P. falciparum*) might have a value of the area between 150 to 1000 pixels. Meanwhile, the other types of the malaria parasite (schizont and gametocyte of *P. falciparum*; ring, mature trophozoite, schizont, and gametocyte of *P. vivax*) might have a value of the area between 3500 to 30,000 pixels. Thus, regions not in these ranges were labeled non-malaria parasites and excluded from the segmented image during the region-growing process to obtain the final image. Here, the analysis of different segmentation techniques was done to find the best segmentation technique that should be used for segmenting thin smear images in all stages of thin blood smear images.

After analyzing different segmentation techniques, a standard segmentation framework was developed using proposed image-processing procedures to segment malaria parasites in all stages of thin blood smear images. The proposed procedures for the segmentation of the malaria parasite, consisting of the best image segmentation techniques determined after the segmentation analysis, were shown in Algorithm 1. The procedures in Algorithm 1 were conducted on ring, mature trophozoite, schizont, and gametocyte images captured from the thin blood smears of *P. falciparum* and *P. vivax* samples. The detailed explanation regarding the proposed image-processing procedures of Algorithm 1 if the malaria parasite had not been segmented properly and a large number of RBCs were still included in these images. Therefore, these types of images should encounter a second segmentation process. These processes include enhancing the malaria image using the MGCS technique with *minp* = 0.05% and *maxp* = 10%, extracting the green component image from the enhanced image, and segmenting the image using the best image segmentation techniques determined after performing the analysis of segmentation previously. 

After a properly segmented malaria image was obtained, the image was filtered using a media filter to remove small background pixels and provide other segmented regions. At this time, the neighborhood of 3 × 3 pixels was chosen to prevent the severe smoothing effect on the segmented image. Afterwards, the SRGAE algorithm was applied to remove the large unwanted regions in the malaria image. When applying the SRGAE algorithm, regions not in the 150–1000 pixels and 3500–30,000 pixels were considered non-malaria parasites and removed from the image during this process. Later, the holes inside the malaria parasite were filled by applying region filling using a 5 × 5 pixels minimum filter. Subsequently, 5 × 5 pixels of erosion process was performed to erode the boundary so that the shape of the malaria parasite was well defined. If the unwanted regions appeared on the image, the image would again be processed using the SRGAE algorithm. By applying the proposed segmentation procedures, a smoother malaria parasite region and a clean segmented malaria image without or less appearance of RBCs region were obtained. The implementations of the proposed procedures for the segmentation of malaria parasite were summarized as follows:
**Algorithm 1: Procedures for segmentation of thin blood smears of *P. falciparum* and *P. vivax*****Input:** Ring, trophozoite, schizont, and gametocyte images**Output:** Segmented images of *P. falciparum* and *P. vivax***Procedures:**Apply the MGCS contrast enhancement technique to the original malaria image.2.Extract the G colour component from the enhanced RGB image.3.Apply the selected segmentation technique that had been determined from the analysis of different segmentation techniques.4.Check the area of background region, *Area[bg]* that satisfies this condition,            a.If the value of *Area[bg]* >= 450,000 pixels, apply the 3×3 pixels median filter.            b.Else, go to step 1.5.Apply the seeded region growing area extraction algorithm.            Note: Regions not in the ranges of 150–1000 and 3500–30,000 were labeled as non-malaria parasites and excluded from the segmented image during this process.6.Apply the 5 × 5 pixels minimum filter.7.Perform the 5 × 5 pixels of the erosion process.8.Check the area of the foreground region, *Area[fg]* that satisfies this condition,            a.If the value of *Area[fg]* >= 9000 pixels, apply the seeded region growing area extraction algorithm
               Note: Regions not in the 3500–30,000 range were labeled as non-malaria parasites and excluded from the segmented image during this process.            b.Colorize the segmented image based on the enhanced RGB image.9.End.

### 3.4. Evaluation of Image Segmentation Quality

In the present study, image segmentation quality was evaluated using three objective indices, accuracy, sensitivity, and specificity. The quality of segmentation of an image was determined based on the pixel similarity of the resultant segmented image against the manually segmented image. The manually segmented image was obtained by manually editing the original malaria image using Adobe Photoshop CS5.1 software, so that the image only consists of the malaria parasite. In contrast, the other regions had been removed from the image. Then, the lab technician validated the visual evaluation of the resultant manual segmented images. In this research, the primary criterion for assessing the performance of image segmentation was based on its ability to produce a fully segmented malaria parasite region and segment the malaria parasite from its complicated blood cell background. The accuracy, sensitivity, and specificity were defined based on Equations (8)–(10), respectively [63].
(8)$$Accuracy=\frac{TP+TN}{TP+TN+FP+FN}\times 100$$
(9)$$Specificity=\frac{TP}{TP+FN}\times 100$$
(10)$$Sensitivity=\frac{TN}{TN+FP}\times 100$$

Consequently, for thin blood smears, the performances of the proposed segmentation method were further evaluated using recall, precision, and F1-score. Precision and recall are used to assess whether the segmented image is over-segmented or under-segmented. A low recall value indicated under-segment, whereas a low precision value indicated over-segment. Here, recall and sensitivity were comparable. A good segmentation would result in an F1-score of 1, which combines precision and recall as a single statistical measure for the segmented image [63].
(11)$$Precision=\frac{TP}{TP+FP}$$
(12)$$Recall=\frac{TP}{TP+FN}$$
(13)$$F1-score=2\times \frac{Precision\times Recall}{Precision+Recall}$$

### 3.5. Feature Extraction for Thick and Thin Blood Smear Images

To classify the malaria parasite or non-malaria parasite successfully, the combination of different types of features, such as size, shape, and color, was used to represent the types of the malaria parasite. For thick smear, a number 32 input features consisting of size, shape, texture, and color-based features had been extracted from the segmented objects, which were given as inputs to the intelligent classifiers. The features used on the segmented thick blood smear images in this study can be classified into four main categories. These features are:Size-based feature: Area, perimeter, solidity, and eccentricity of the segmented object.Shape-based feature: Hu’s moment invariant of the segmented object.Texture-based feature: Haralick and Hausdorff dimension.Color-based feature: Mean and standard deviation of the segmented object’s red, green, and blue color components.

Meanwhile, for thin smear, 28 input features consisting of size, shape, and color-based features were extracted from the segmented objects and were given as inputs to the intelligent classifiers. The features used in this study’s segmented thin blood smear images can be classified into three main categories. These features are:Size-based feature: area of the segmented object.Shape-based feature: Second and third order central moment, affine invariant moment, and Hu’s moment invariant of the segmented object.Color-based feature: Mean and variance of red, green, blue, intensity, and saturation color components of the segmented object.

### 3.6. Malaria Diagnosis Using Intelligent Classifier

After the features had been extracted, the suitability of utilizing these features for the classification of the thick and thin smear blood samples was tested using three different classifiers. These three different classifiers were single-hidden layer feed-forward neural networks (SLFN) trained by ELM and OS-ELM, and RFs. The classifications of the individual segmented object (malaria parasite and non-malaria parasite) inside the thick and thin smear blood samples using the three intelligent classifiers consist of one task in thick smear blood samples and five tasks in thin smear blood samples. As for thick smear, the classification task was to classify malaria and non-malaria parasites. Meanwhile, five tasks in thin smear are:Task 1: Classification between the malaria parasite and non-malaria parasite.Task 2: Classification of malaria parasite as either *P. falciparum* or *P. vivax* species.Task 3: Classification of *P. falciparum* species into the ring, mature trophozoite, schizont, and gametocyte stages.Task 4: Classification of *P. vivax* species into the ring, mature trophozoite, schizont, and gametocyte stages.Task 5: Classification of the individual segmented object inside the malaria sample into eight categories, such as *P. falciparum* (Ring, Trophozoite, Schizont, and Gametocyte) and *P. vivax* (Ring, Trophozoite, Schizont, and Gametocyte).

Thick blood smear training and validation data distributions were tabulated in Table 1, whereas thin blood smear training, validation, and testing data distributions were tabulated in Table 2. In an attempt to classify the malaria parasite and non-malaria parasite inside the thick blood smear sample, a number of 7500 segmented objects were obtained from 300 malaria images. A total of 4500 segmented objects were used as training data, and 3000 were used as validation data. Meanwhile, in a thin blood smear sample, a number of 6300 segmented objects had been obtained from 4035 malaria images. A total of 3380 segmented objects were used as training data, while 1260 were used as validation and testing data. 

#### 3.6.1. Classification Using Extreme Learning Machine (ELM)

In this study, the ELM classification process for training the SLFN proposed by Huang et al. [63] was used to classify the target in thick and thin blood smear images. ELM is a supervised batch learning algorithm that employs random computational nodes in the hidden layer and computes its output weights analytically by solving a general linear system equation. Basically, ELM is made of three layers: input, hidden, and output. Here, a sigmoid activation function was used for both hidden and output nodes. Note that these activation functions can effectively avoid the insufficient expression ability of neural networks caused by linear mapping. During the ELM training process, the analysis of finding the best number of states and the best number of hidden nodes was conducted to obtain the best classification result. Here, the number of hidden nodes was optimized by varying the hidden nodes from 1 to 50 with the step of 1. The state is a seed that can be used to initialize weights to get consistent results. As for the analysis of states, the number of states varied between the value of 1 and 4 to optimize the classification performance. The full implementation of ELM for training an SLFN could be referred to in [64]. 

#### 3.6.2. Classification Using Online Sequential Extreme Learning Machine (OS-ELM)

In this study, the next classification process was conducted using an SLFN trained by OS-ELM, which was proposed by Liang et al. [47]. OS-ELM is an upgraded online sequential version of the ELM. As for the OS-ELM, this algorithm can learn the training data one-by-one and chunk-by-chunk, discarding the data for which training had already been done [47]. Similar to ELM, the analysis of finding the best number of states and hidden nodes was conducted to obtain the best classification result. However, in sequential learning, only one epoch was needed, and the training stopped right after all the training patterns were processed. In terms of performance, the OS-ELM has been proven to be faster than the other well-known sequential algorithms and produces better generalization performance with lower training time. The full implementation of OS-ELM for training an SLFN could be referred to in Liang et al. [47].

#### 3.6.3. Classification Using Random Forest

RF is one of the potential algorithms for building classifiers introduced by Breiman [65]. This classifier consists of many binary decision trees, where each tree is a classifier by itself, given a certain weight for its classification output. A binary decision tree is a method of using nodes in a tree structure to test the attributes of a dataset. The results of these tests were used to split the training data into subsets, which were then passed onto the next layer of the tree. This continues until each subset at a node contains only one class. As an ensemble of models, the RF used majority voting for the classification task and averaging for regression to make a final prediction [65]. For classification using RF, the decision tree algorithm used during the classification process is Classification and Regression Tree (CART) [65]. In order to obtain the best classification result in this study, the optimization parameter for the number of trees in the forest was analyzed. Therefore, the number of trees would vary from 1 to 50 to optimize this parameter. Note that the procedures for finding the best number of trees were applied for each classification task.

## 4. Results and Discussion

This section provides the results of segmentation performance using various segmentation techniques and the result of classification performance of thick and thin blood smears using three different intelligent classifiers. Here, a total of 11 segmentation techniques: adaptive threshold, Niblack, Sauvola, Phansalkar, Nick, Bradley, Feng, KM, FKM, EKM, and FCM have been used on thick and thin blood smear images of *P. falciparum* and *P. vivax* species to obtain the best segmented resultant image as can be seen from Figure 5, Figure 6 and Figure 7, respectively. 

The comparison of the image segmentation techniques for thick and thin blood smear images is presented in Table 3, and the resultant images have been displayed in Figure 5, Figure 6 and Figure 7, respectively. Based on the findings summarized in Table 3, the Feng technique is unsuitable for thick smear images since this technique is unable to segment malaria parasites properly, yet produces over-segment and resultant noisy images. In addition, Sauvola, KM, FKM, and EKM techniques are able to segment the thick smear image, but in some cases, the parasites become thin and disappear. Thus, the suitable image segmentation technique for thick blood smear images is the Phansalkar technique, as this technique can segment the thick smear image properly. As for thin smear, all techniques can segment thin smear images, but some techniques, such as Niblack, Sauvola, Nick, and Feng techniques, produce over- and under-segment resultant images and noisy images, and, in some cases, the parasites disappear. Additionally, clustering techniques, such as KM, FKM, and FCM, produce over-segment and resultant noisy images. Note that the Phansalkar technique is unsuitable for thin smears because this technique produces under-segment and resultant noisy images. Based on the findings, the suitable image segmentation technique for thin blood smears is the EKM clustering technique. EKM technique is able to segment thin smear images and performs well by producing a clean segmented image.

### 4.1. Analysis of Thick Smear Segmentation 

In order to evaluate the proposed image segmentation procedure and select the best segmentation technique for thick smear images, the performance of all 11 segmentation techniques is further analyzed by comparing the resultant segmented image against the manually segmented image. Table 4 presents the results of segmentation performances based on accuracy, specificity, and sensitivity evaluation obtained from the average of 200 thick blood smear images. The best results obtained for analyses using different segmentation techniques are made bold. Specificity evaluation indicates the percentage of actual negative values was correctly identified. Meanwhile, sensitivity evaluation indicates the percentage of actual positive values was correctly identified. Based on the average of segmentation performances in Table 4, for the segmentation accuracy, segmentation using the FKM clustering has proven to be good in segmenting the entire region in thick blood smear images compared to other segmentation techniques with the highest accuracy of 99.91%. Nevertheless, even though the accuracy of the FKM technique is highest compared to other segmentation techniques, FKM cannot obtain a fully segmented malaria parasite region, as seen in Figure 6 previously. 

Based on the segmentation performance of specificity evaluation in Table 4, segmentation using adaptive thresholding has proven to produce a clean segmented image compared to other segmentation techniques, with the highest specificity of 99.96%. However, even though the adaptive thresholding technique has a high specificity value, this technique produces a lower sensitivity value compared to other segmentation techniques with a value of 40.04%, which means that this technique cannot obtain a fully segmented malaria parasite region. The main element used to evaluate the segmentation performance is based on its capability to entire segment region in thick blood smear image, including obtaining the fully segmented malaria parasite region and producing a clean segmented resultant image. Therefore, considering this element, segmentation using the Phansalkar technique has the best performance by providing the highest sensitivity results with a value of 92.47% compared to the other segmentation techniques. It can also achieve high accuracy and specificity with a value of 99.86% and 99.87%, respectively.

### 4.2. Analysis of Thin Smear Segmentation

In order to evaluate the proposed image segmentation procedure and select the best segmentation technique for thin smear images, the performance of all 11 segmentation techniques is further analyzed by comparing the resultant segmented image against the manually segmented image. Table 5 presents the results of segmentation performances based on accuracy, specificity, sensitivity, precision, recall, and F1-score evaluation obtained from the average of 100 thin blood smear images of a ring, trophozoite, schizont, and gametocyte stages. In essence, the primary factor considered to assess the segmentation performance is based on its ability to segment the entire region in a thin blood smear image, including obtaining the fully segmented malaria parasite region and the ability to produce a clean segmented resulting image. 

Based on the average of segmentation performances in Table 5, for segmentation accuracy, segmentation using the EKM clustering has proven to be good in segmenting the entire region in thin blood smear images compared to other segmentation techniques with the highest accuracy value of 99.2%. Subsequently, based on the segmentation performance of specificity evaluation in Table 5, segmentation using the nick technique has proven to produce a clean segmented image compared to other segmentation techniques with the highest specificity value of 99.92%. However, even though the Nick technique has a high specificity value, this technique produces a low recall value of 0.5895. Therefore, this technique has produced an under-segmented resultant image, as seen in Figure 7. Apart from that, based on the segmentation performance of sensitivity evaluation in Table 5, segmentation using the FKM clustering technique has proven to be good in obtaining the fully segmented malaria parasite region compared to other segmentation techniques, with the highest sensitivity value of 93.87%. Nevertheless, even though the FKM clustering technique has a high specificity value, this technique produces a low precision value with a value of 0.5103 means that this technique has produced an over-segmented resultant image. 

By considering the primary factor that has been mentioned previously, segmentation using the EKM clustering technique has the best performance. It provides the highest accuracy results, meaning that this technique can segment the entire region of a thin blood smear image with a value of 99.2% and the highest F1-score. Furthermore, it means that the EKM clustering technique is able to obtain a good, segmented image with a value of 0.9033 compared to the other segmentation techniques. Additionally, this technique can achieve good specificity, sensitivity, precision, and recall value of 99.58%, 87.52%, 0.9579, and 0.8752, respectively. Furthermore, as seen in Figure 7, these resultant images show that using the proposed image segmentation procedure of the EKM clustering technique has produced good segmented thin blood smear images consisting of *P. falciparum* and *P. vivax* species in all stages.

The analysis based on the proposed segmentation framework as presented in Algorithm 1 has also been conducted using 400 (50 PF_Ring, 50 PF_Trophozoite, 50 PF_Schizont, 50 PF_Gametocyte, 50 PV_Ring, 50 PV_Trophozoite, 50 PV_Schizont, and 50 PV_Gametocyte) images from different malaria blood slides. Figure 8 and Figure 9 provide the results of the proposed segmentation procedures applied to malaria images of the ring, mature trophozoite, schizont, and gametocyte stages captured from *P. falciparum* and *P. vivax* blood samples. In addition, Table 6 tabulates the segmentation performance based on sensitivity, specificity, and accuracy for the final segmented images of eight different types of malaria. Here, the final segmented images’ sensitivity, specificity, and accuracy are determined based on the pixel similarity of the final segmented image against the manually segmented image. Overall, the proposed segmentation method has successfully segmented 400 malaria images with segmentation accuracy, sensitivity, and specificity of 99.49%, 84.07%, and 99.77%, respectively.

### 4.3. Analysis of Thick Smear Classification 

The findings of employing ELM, OS-ELM, and RF intelligent classifiers to distinguish between malaria parasites and non-malaria parasites inside thick blood samples are presented in this section. Here, the result of the classification between the malaria parasite and non-malaria parasite inside the thick blood samples will start with the analysis using ELM. Followed by classification using OS-ELM and ended with the classification using RF. The first classification process is to classify the malaria parasites and non-malaria parasites using ELM. Note that the analysis of the number of hidden nodes for classification between the malaria parasites and non-malaria parasites using ELM is shown in Table 7. According to the results in Table 7, ELM can detect malaria parasites with a training accuracy of 83%, and the best validation accuracy of 83.83%. 

Next, the second classification process to classify malaria and non-malaria parasites use OS-ELM. The analysis of the number of hidden nodes for classification between the malaria parasites and non-malaria parasites using OS-ELM is shown in Table 7. It has been illustrated in Figure 10a. According to the values highlighted bold in Table 7, OS-ELM effectively detects malaria parasites, with a training accuracy of 85.37% and the best validation accuracy of 86.39%. RF is the third intelligent classifier to classify malaria and non-malaria parasites. Table 7 and Figure 10b demonstrate the analysis of the number of trees for classification between the malaria parasites and non-malaria parasites using RFs. The results present that the RF can detect the malaria parasite by producing the best classification result with high training and validation accuracies of 95.44% and 86.89%, respectively. 

Here, the classification performance of three intelligent classifiers for identifying malaria and non-malaria parasites is summarized in Table 7. According to the data tabulated in Table 7, all these intelligent classifiers of ELM, OS-ELM, and RF network have exhibited good classification performance by achieving training and validation accuracy values higher than 80%. By comparing the classification results between the three intelligent classifiers, the RF network has proven to be the best by producing the highest validation accuracies of 86.89% for classifying malaria and non-malaria parasites. Consequently, it is followed by the classification results provided by OS-ELM in the second and ELM in the third place, with values of 86.39% and 83.83%, respectively. 

### 4.4. Analysis of Thin Smear Classification 

This section provides the results of classifying the individual segmented object (the malaria parasite and non-malaria parasite) inside the thin blood samples using ELM, OS-ELM, and RF intelligent classifiers. Here, the classifications of individual segmented objects involve five main tasks previously described in Section 3. For each category of the classification task, the result and discussion will start with the analysis using ELM, followed by classification using OS-ELM, and end with the classification using RF. The following sub-sections will discuss the results obtained after performing the analysis using the three intelligent classifiers. 

#### 4.4.1. Classification Results of Malaria Parasites Detection

In this section, the three different classifiers have been applied to perform the classification process based on Task 1. This task involves the classification between the malaria parasite and the non-malaria parasite. The first classification process to perform Task 1 is the classification using ELM. Table 8 summarizes the classification performances between the malaria parasite and non-malaria parasite using ELM. Based on the results in Table 8, ELM is also capable of detecting malaria parasites with good generalization properties by producing good training accuracy of 94.29%, and producing the best validation and testing accuracies of 93.57% and 93.81%, respectively. Consequently, the second classification process to perform Task 1 is using OS-ELM. Figure 11a illustrates the analysis of several hidden nodes for classifying the malaria parasite and non-malaria parasite using OS-ELM. Meanwhile, Table 8 summarizes the performances of classification between malaria parasites and non-malaria parasites using OS-ELM. Based on the results in both tables, OS-ELM can detect malaria parasites with good generalization properties by producing good training accuracy of 94.29% and the best validation and testing accuracies of 93.57% and 93.81%, respectively. Apart from that, the third classification process to perform Task 1 is the classification using RF. Table 8 and Figure 11b demonstrate the analysis of the number of trees for classifying malaria parasites and non-malaria parasites using RFs. The results indicate that the RF can detect malaria parasites with good generalization properties by producing the best classification result with training, validation, and testing accuracies of 97.80%, 94.37%, and 94.29%, respectively.

#### 4.4.2. Classification Results of Malaria Parasites Species

This section provides the classification results based on Task 2, which involves classifying *P. falciparum* and *P. vivax* species. The first classification process to perform Task 2 is the classification using ELM. Table 8 summarizes the performances of classification between malaria parasites and non-malaria parasites using ELM, OS-ELM, and RF. Based on the results in Table 8, ELM can differentiate between *P. falciparum* and *P. vivax* species with good training accuracy of 94.29%, and producing the best validation and testing accuracies of 93.57% and 93.81%, respectively. Subsequently, the second classification process to perform Task 2 is using OS-ELM. Based on the results in Table 8, OS-ELM can differentiate between *P. falciparum* and *P. vivax* species by producing a good training accuracy of 97.68%, and the best validation and testing accuracies of 97.55% and 97.06%, respectively. Apart from that, the third classification process to perform Task 2 is the classification using RF. The results from Table 8 exhibit that the RF has the capability to differentiate between *P. falciparum* and *P. vivax* species by producing training, validation, and testing accuracies of 99.28%, 98.82%, and 93.73%, respectively.

#### 4.4.3. Classification Results of Malaria Parasites Life-Cycle Stages

This section provides the classification results based on Tasks 3 and 4, which involve the classifications of *P. falciparum* and *P. vivax* species into their four life-cycle stages: ring, mature trophozoite, schizont, and gametocyte stages. The classification using ELM is the first classification process to perform Tasks 3 and 4. Table 8 summarizes the performances of classification among ring, mature trophozoite, schizont, and gametocyte stages of *P. falciparum* and *P. vivax* species using ELM. Based on these results, the ELM can classify the four life-cycle stages of *P. falciparum* with high validation and testing accuracies compared to *P. vivax* species. Here, the validation accuracy of 97.24%, and testing accuracy of 94.14% that ELM has produced for the recognition of *P. falciparum* species are higher as compared to the recognition of *P. vivax* species with validation and testing accuracies of 86.36% and 77.95%, respectively. Correspondingly, the second classification process to perform Tasks 3 and 4 is the classification utilizing OS-ELM. The analysis of the number of hidden nodes for classification among ring, mature trophozoite, schizont, and gametocyte stages of *P. falciparum* and *P. vivax* species using OS-ELM are present in Figure 12a and Figure 13a and summarized in Table 8. Based on these results, the OS-ELM can classify the four life-cycle stages of *P. falciparum* with high validation and testing accuracies compared to *P. vivax* species. The validation accuracy of 97.24%, and testing accuracy of 94.14% that OS-ELM has produced for the recognition of *P. falciparum* species are higher as compared to the recognition of *P. vivax* species with validation and testing accuracies of 86.36% and 77.95%, respectively. In addition, the third classification process to perform Tasks 3 and 4 is RF classification. The analysis of the number of trees for the classification of different life-cycle stages in *P. falciparum* and *P. vivax* species employing RF is demonstrated in Figure 12b and Figure 13b and summarized in Table 8. These results indicate that the diagnosis among different stages in *P. falciparum* species with validation accuracy of 98.79% and testing accuracy of 96.38% is easier to recognize and differentiate than the *P. vivax* species with validation and testing accuracies of 84.77% and 80%, respectively.

#### 4.4.4. Classification Results of Multiclass Classification for Malaria Parasites Species and Life-Cycle Stages

This section provides the multiclass classification results based on Tasks 5. Here, Task 5 involves classifying the individual segmented object inside the malaria sample into eight classes which are PF_Ring, PF_Trophozoite, PF_Schizont, PF_Gametocyte, PV_Ring, PV_Trophozoite, PV_Schizont, and PV_Gametocyte stages. The first classification process to perform Task 5 is the classification using ELM. Table 8 summarizes the performances of classifying the malaria parasite into eight classes using ELM. Based on these results, the ELM has the ability to classify the individual segmented object inside the malaria sample into eight classes of PF_Ring, PF_Trophozoite, PF_Schizont, PF_Gametocyte, PV_Ring, PV_Trophozoite, PV_Schizont, and PV_Gametocyte stages. Here, the validation accuracy of 97.24% and testing accuracy of 94.14% have been produced by ELM for the classification of malaria parasites into eight classes. Subsequently, the second classification process to perform Task 5 is the classification using OS-ELM. The analysis of the number of hidden nodes for classifying malaria parasites into eight classes using OS-ELM is summarized in Table 8. Here, the validation accuracy of 89.12%, and testing accuracy of 86.86%that have been produced by OS-ELM for the classifications of the individual segmented object inside the malaria sample into eight classes of PF_Ring, PF_Trophozoite, PF_Schizont, PF_Gametocyte, PV_Ring, PV_Trophozoite, PV_Schizont, and PV_Gametocyte stages. In addition, the third classification process to perform Task 5 is the classification using RF. Note that the analysis of the number of trees for classifications of the individual segmented object inside the malaria sample into eight classes which are PF_Ring, PF_Trophozoite, PF_Schizont, PF_Gametocyte, PV_Ring, PV_Trophozoite, PV_Schizont, and PV_Gametocyte stages are summarized in Table 8. The classification results show that the RF can classify malaria parasites into eight classes by producing good results in the validation and testing phases with accuracies of 90.78% and 82.25%, respectively. 

Table 8 presents the classification results for the five different tasks produced by the three intelligent classifiers. The best results obtained for analyses using these classifiers are made bold. Based on the results in Table 8, the three intelligent classifiers provide good classification performance with training, validation, and testing accuracies of more than 80%, except for the classification of Task 4 using OS-ELM. In addition, RF produces the best classification results for both the validation and testing phases for Tasks 1 and 3. As for the classification of Tasks 2 and 5, the RF network produces the best validation accuracy, while the best testing accuracy has been produced by OS-ELM. However, for the classification of Task 4, the OS-ELM network produces the best validation accuracy, but the best testing accuracy has been produced by RF. By comparing the classification results of the five tasks provided by the three intelligent classifiers, the RF network has proven to be the best by producing high validation accuracies for Tasks 1, 2, 3, and 5, and high testing accuracies for Tasks 1, 3, and 4. Although RFs obtain better accuracy than ELM and OS-ELM, this classifier requires a lot of training time and is quite slow to make predictions. Even though the ELM and OS-ELM algorithms are computationally efficient and relatively faster, the robustness and stability of both ELM and OS-ELM algorithms cannot always be guaranteed. This is because the random generation of the parameters for the hidden nodes will result as the model’s expected output may not match the actual output. Hence, due to good classification performance, the RF network has been selected to classify blood images from both normal and malaria samples. 

## 5. Discussion

A rapid and accurate malaria diagnosis is essential as a precautionary measure to prevent more malaria infections. Thus, this study presents a novel multi-stage malaria parasite recognition framework for microscopy diagnosis of both thick and thin smear images. This study aims to deal with parasite detection in thick smear microscopy, and parasite species classification and staging for thin smear microscopy. In brief, Table 9 presents the classifier performance reported in the recent literature related to malaria diagnosis, which is comparable to the performance obtained from this study. However, certain noteworthy elements that are distinct from other recent works are presented in this study. First, most of the relevant studies utilized thin blood smear samples for parasite detection, as in [25,29,37,38,39,40,41,42,43,44]. Based on clinical microscopy tests, a thick blood smear sample is more suitable for performing parasite detection since it is more sensitive to the parasite.

In contrast, a thin blood smear sample is suitable for identifying species and life-cycle stages. Yet, this study applies both thick and thin smear images for malaria parasite recognition. Second, the malaria diagnosis techniques, based on the standard machine learning procedure as in [25,27,29,33,36], use small datasets. The majority of them employ datasets with fewer than 300 total images. Nonetheless, adding more data will help in producing a better result. Therefore, to achieve superior performance, this study used a total of 500 thick blood smear images and 4035 thin blood smear images, which include both *P. falciparum* and *P. vivax* species. Based on Table 9, most recent studies used deep learning [37,38,39,40,41,42,43,44] and achieved high accuracies. However, deep learning models require a huge amount of data to learn and make an effective prediction. For instance, [34,35,37,38,39,40,41,42,43,44] utilized about 27,558 images as datasets. Nevertheless, even though our proposed work used standard machine learning procedures, our proposed work also demonstrates higher accuracies in parasite detection on thick and thin blood smear samples. This includes high accuracies in species recognition and staging identification with the value of 86.89%, 94.37%, 98.82%, and 90.78%, respectively. The values are higher than other researchers that also used machine learning, such as in [25] and [36]. However, it is also comparable to the researcher that used deep learning, such as in [41,43]. Therefore, the results presented in this study are comparable to the state-of-the-art results but also reliable and generalizable. Note that it was trained and validated on a quite large dataset compared to other recent machine learning works.

## 6. Conclusions

In this study, a full framework based on image processing and machine learning was designed to address the problem of detecting parasites for quantifying parasitemia levels in thick smear microscopy, classifying parasite species, and staging for thin smear microscopy. A total of 11 segmentation techniques, namely, adaptive thresholding, Bradley, Niblack, Nick, Sauvola, Phansalkar, Feng, KM, FKM, EKM, and FCM clustering have been employed to explore the effect of segmentation on thick and thin smear images. The proposed image-processing framework for thick smears using the Phansalkar segmentation technique has established the highest potential for segmenting the entire region of a thick blood smear image. This includes obtaining the fully segmented malaria parasite region and producing a clean segmented resultant image by reaching an overall accuracy of 99.8593%, specificity of 99.8678%, and sensitivity of 92.4705%. Meanwhile, the proposed image-processing framework for thin smears using the EKM clustering technique has established the highest potential for segmenting the entire region of thin blood smear images. This includes obtaining the fully segmented malaria parasite region and producing a clean segmented resultant image by achieving the best accuracy, specificity, sensitivity, precision, recall, and F1-score value of 99.19932%, 99.57775%, 87.5209%, 0.957919, 0.875209, and 0.90334, respectively. Overall, the proposed intelligent diagnostic system for malaria has performed the detection process of thick smear images with a classification accuracy of 86.39% using an RF classifier. As for the diagnosis process, the system based on the RF has correctly classified *P. falciparum* and *P. vivax* species with 98.82% accuracy and a staging result of 90.78%. Combining these approaches will allow for versatile detection and stage categorization of malaria parasites from thick and thin smear images with an interactive assessment of the results. Hence, it will improve inspection reproducibility and present a standard lab routine lab for future field-based automated malaria diagnosis. This study focused on processing the malaria images from only two malaria species, which are *P. falciparum* and *P. vivax* species. Therefore, to improve the research performance for future work, diagnosis of malaria disease based on all others malaria species, such as *P. ovale*, *P. malariae,* and *P. knowlesi* species, could be considered so that this system could be used worldwide.

## Figures and Tables

**Figure 1 diagnostics-13-00511-f001:**
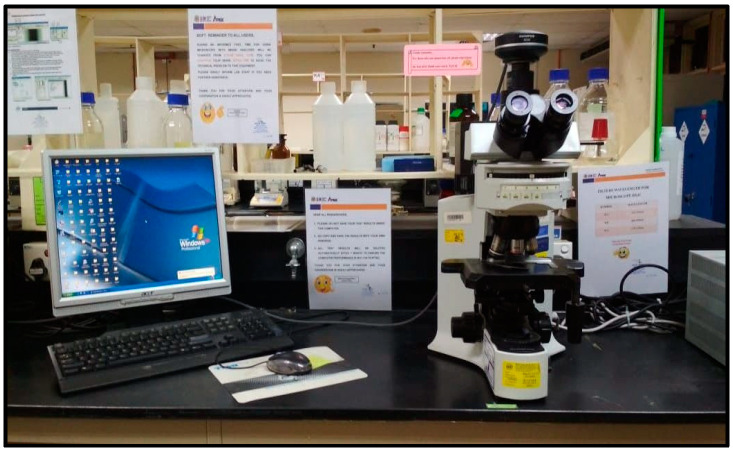
*OLYMPUS BX41* digital microscope.

**Figure 2 diagnostics-13-00511-f002:**
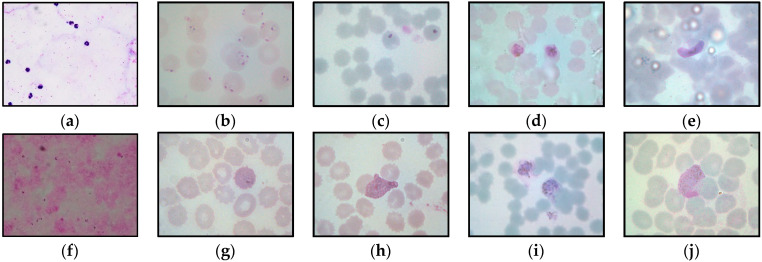
The samples of the captured blood images consisting of (**a**) *P. falciparum* of thick smear; (**b**–**e**) *P. falciparum* of thin smear in the ring, trophozoite, schizont, gametocyte; (**f**) *P. vivax* of thick smear; (**g**–**j**) *P. vivax* of thin smear in the ring, mature trophozoite, schizont, gametocyte.

**Figure 3 diagnostics-13-00511-f003:**
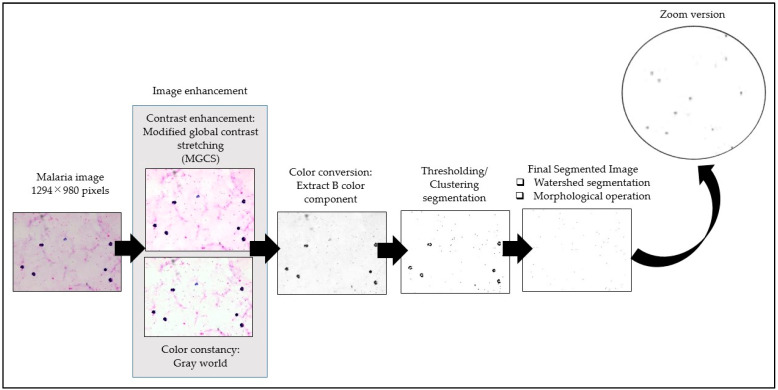
The framework of the proposed image-processing procedure for thick smear segmentation.

**Figure 4 diagnostics-13-00511-f004:**
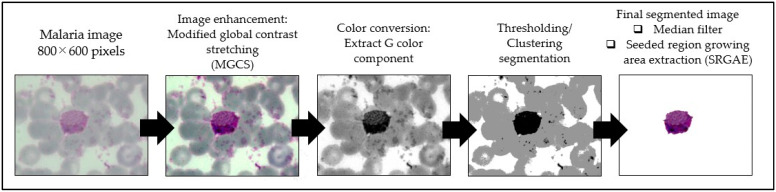
The framework of the proposed image-processing procedure for thin smear segmentation.

**Figure 5 diagnostics-13-00511-f005:**
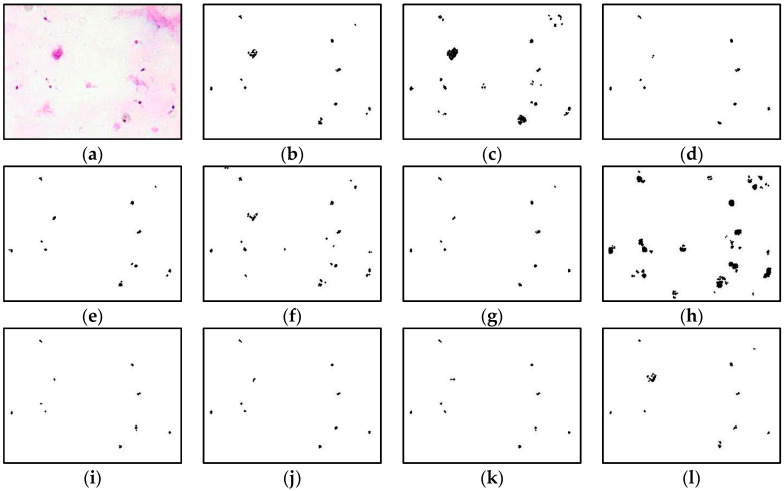
*P. falciparum* of thick smear in various segmentation techniques of (**a**) original image; (**b**) adaptive thresholding; (**c**) Niblack; (**d**) Sauvola; (**e**) Phansalkar; (**f**) Bradley; (**g**) Nick; (**h**) Feng; (**i**) KM; (**j**) FCM; (**k**) FKM; (**l**) EKM.

**Figure 6 diagnostics-13-00511-f006:**
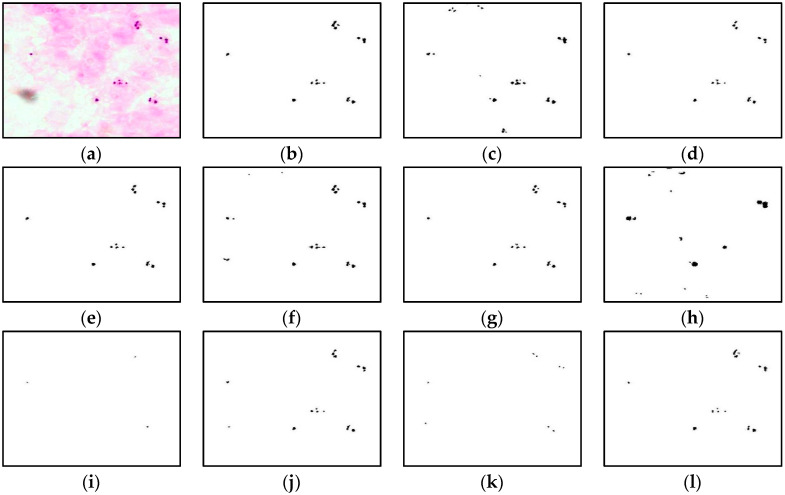
*P. vivax* of thick smear in various segmentation techniques of (**a**) original image; (**b**) adaptive thresholding; (**c**) Niblack; (**d**) Sauvola; (**e**) Phansalkar; (**f**) Bradley; (**g**) Nick; (**h**) Feng; (**i**) KM; (**j**) FCM; (**k**) FKM; (**l**) EKM.

**Figure 7 diagnostics-13-00511-f007:**
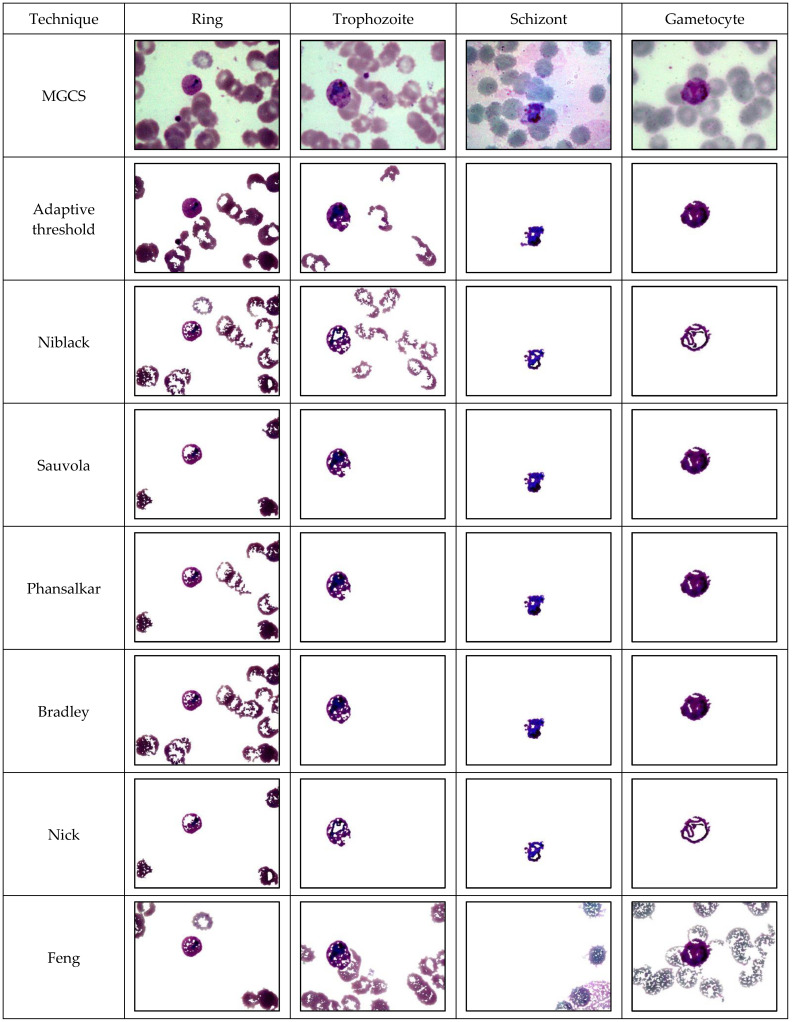
*P. vivax* of thin smear in the ring, mature trophozoite, schizont, and gametocyte in various segmentation techniques.

**Figure 8 diagnostics-13-00511-f008:**
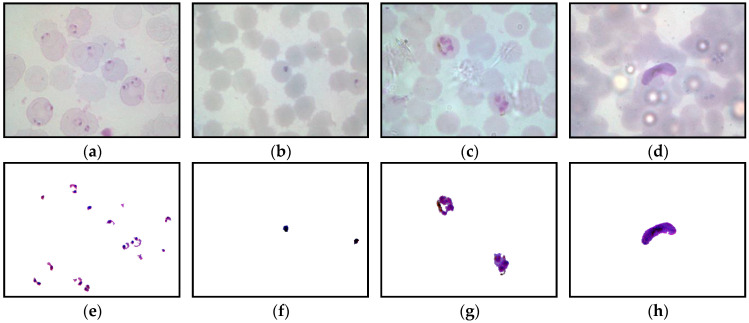
Original and final segmented images for *P. falciparum*: (**a**) Original ring; (**b**) Original trophozoite; (**c**) Original schizont; (**d**) Original gametocyte; (**e**) Segmented ring (**f**) Segmented trophozoite; (**g**) Segmented schizont; (**h**) Segmented gametocyte.

**Figure 9 diagnostics-13-00511-f009:**
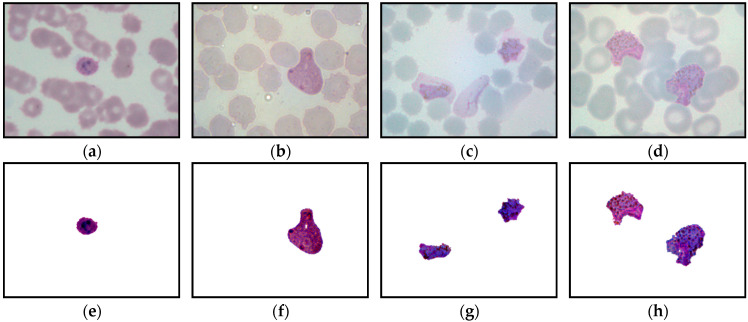
Original and final segmented images for *P. vivax*: (**a**) Original ring; (**b**) Original trophozoite; (**c**) Original schizont; (**d**) Original gametocyte; (**e**) Segmented ring (**f**) Segmented trophozoite; (**g**) Segmented schizont; (**h**) Segmented gametocyte.

**Figure 10 diagnostics-13-00511-f010:**
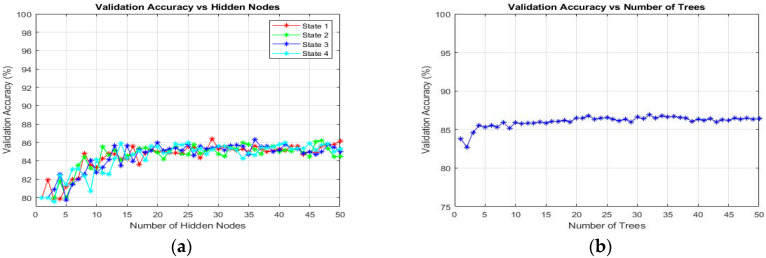
Analysis for classification between malaria and non-malaria parasites: (**a**) OS-ELM; (**b**) Random forest.

**Figure 11 diagnostics-13-00511-f011:**
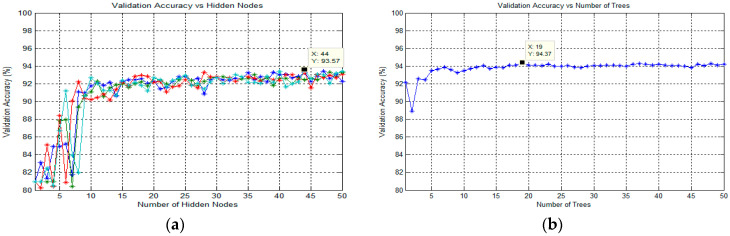
Analysis for classification between the malaria parasite and non-malaria parasite: (**a**) OS-ELM; (**b**) Random forest.

**Figure 12 diagnostics-13-00511-f012:**
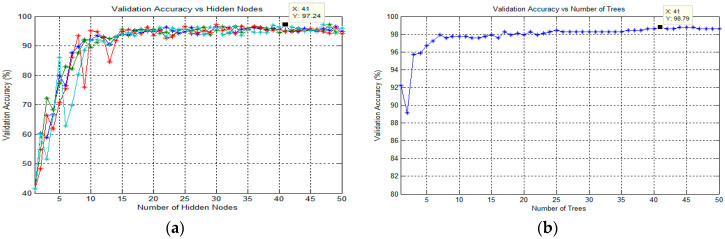
Analysis for classifications of *P. falciparum* species into their four life-cycle stages: ring, mature trophozoite, schizont, and gametocyte stages: (**a**) OS-ELM; (**b**) Random forest.

**Figure 13 diagnostics-13-00511-f013:**
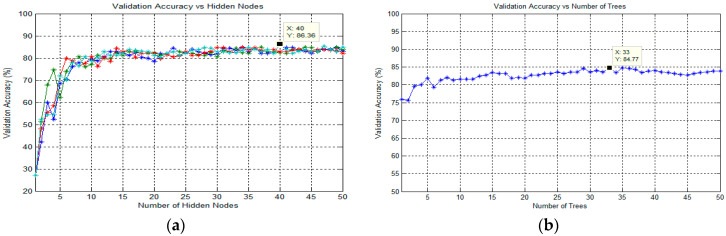
Analysis for classifications of *P. vivax* species into their four life-cycle stages: ring, mature trophozoite, schizont, and gametocyte stages: (**a**) OS-ELM; (**b**) Random forest.

**Table 1 diagnostics-13-00511-t001:** Training and validation data distributions for the thick smear.

Categories	Training Data	Validation Data
Non-malaria parasite	1000	1000
PF	1750	1000
PV	1750	1000
Total Data	4500	3000

**Table 2 diagnostics-13-00511-t002:** Training, validation, and testing data distributions for the thin smear.

Categories	Training Data	Validation Data	Testing Data
Non-malaria parasite	720	240	240
PF_Ring	720	240	240
PF_Trophozoite	300	100	100
PF_Schizont	360	120	120
PF_Gametocyte	360	120	120
PV_Ring	300	100	100
PV_Trophozoite	300	100	100
PV_Schizont	360	120	120
PV_Gametocyte	360	120	120
Total Data	3780	1260	1260

**Table 3 diagnostics-13-00511-t003:** Findings of investigated segmentation techniques on thick and thin blood smear images.

Segmentation Method	Findings on Thick Smear (Refer to Figure 5 and Figure 6)	Findings on Thin Smear (Refer to Figure 7)
Adaptive threshold	This technique is able to segment thick smear images, but in some cases, this technique will produce a noisy image.	This technique is able to segment thin smear images, but in some cases, it produces an over-segment and noisy image.
Niblack	This technique is able to segment thick smear images, yet this technique will produce an over-segment and noisy image.	This technique is able to segment thin smear images. However, it produces both over- and under-segment, and noisy images. In some cases, the parasites disappear.
Sauvola	This technique is able to segment thick smear images. However, in some cases, the parasites become thin and disappear.	This technique can segment thin smear images but produces under-segment and noisy images. Plus, in some cases, the parasites disappear.
Phansalkar	This technique is able to segment thick smear images and perform better, especially on images with uneven lighting conditions.	This technique is able to segment thin smear images. However, in some cases, it produces an under-segment and noisy image.
Bradley	This technique is able to segment thick smear images, but in some cases, it produces an over-segment and noisy image.	This technique is able to segment thin smear images, but in some cases, it produces an under-segment and noisy image.
Nick	This technique is able to segment thick smear images; however, in some cases, the parasites disappear.	This technique is able to segment thin smear images but produces under-segment, noisy images, causing some parasites to disappear.
Feng	This technique is unable to segment thick smear images as it produces over-segment and noisy images.	This technique is able to segment thin smear images. However, it produces both over and under-segments and noisy images; in some cases, the parasites disappear.
KM	This technique is able to segment thick smear images. However, in some *P. falciparum* images, the parasites become thin and disappear.	This technique is able to segment thin smear images. However, in some cases, it produces over-segment and noisy images.
FCM	This technique is able to segment thick smear images, but in some cases, it produces over-segment and noisy images.	This technique is able to segment thin smear images. However, in some cases, it produces an over-segment and noisy image.
FKM	This technique is able to segment thick smear images. However, in some *P. falciparum* images, the parasites become thin and disappear.	This technique is able to segment thin smear images. However, in some cases, it produces an over-segment and noisy image.
EKM	This technique is able to segment thick smear images. However, in some *P. falciparum* images, the parasites become thin and disappear.	This technique is able to segment thin smear images and perform better by producing a clean segmented image.

**Table 4 diagnostics-13-00511-t004:** Average segmentation evaluation consisting of accuracy, specificity, and sensitivity of 200 thick blood smear images.

Techniques	Accuracy (%)	Specificity (%)	Sensitivity (%)
Adaptive thresholding	99.89	99.96	40.04
Bradley	99.63	99.64	76.71
Feng	98.70	98.75	52.35
Niblack	99.81	99.82	83.85
Nick	99.83	99.84	88.02
**Phansalkar**	**99.86**	**99.87**	**92.47**
Sauvola	99.89	99.91	81.71
*k*-means	99.91	99.94	70.32
Fuzzy *c*-means	98.77	98.82	57.44
Fast *k*-means	99.91	99.93	75.75
Enhanced *k*-means	99.86	99.90	74.43

**Table 5 diagnostics-13-00511-t005:** Average segmentation evaluation consisting of accuracy, specificity, sensitivity precision, recall, and F1-score of 100 thin blood smear images.

Techniques	Accuracy (%)	Specificity (%)	Sensitivity (%)	Precision	Recall	F1-Score
Adaptive thresholding	97.40	97.75	87.15	0.7914	0.8716	0.7870
Bradley	98.81	99.33	83.71	0.9011	0.8372	0.8525
Feng	80.20	81.49	43.98	0.0670	0.4398	0.2227
Niblack	94.65	95.70	63.81	0.5867	0.6381	0.5539
Nick	98.58	99.92	58.95	0.9715	0.5895	0.7598
Phansalkar	99.18	99.74	83.13	0.9457	0.8313	0.8783
Sauvola	99.28	99.89	81.59	0.9672	0.8159	0.8882
*k*-means	95.87	96.11	88.63	0.6867	0.8863	0.7099
Fuzzy *c*-means	97.95	98.25	89.43	0.7821	0.8943	0.8010
Fast *k*-means	93.69	93.72	93.87	0.5103	0.9387	0.5870
**Enhanced *k*-means**	**99.20**	**99.58**	**87.52**	**0.9579**	**0.8752**	**0.9033**

**Table 6 diagnostics-13-00511-t006:** Segmentation performances for the final segmented malaria images.

Images	Sensitivity (%)	Specificity (%)	Accuracy (%)
PF_Ring	75.95	99.56	99.35
PF_Trophozoite	76.22	99.86	99.81
PF_Schizont	82.23	99.95	99.68
PF_Gametocyte	88.35	99.90	99.68
PV_Ring	88.87	99.22	98.92
PV_Trophozoite	91.59	99.95	99.65
PV_Schizont	81.27	99.88	99.50
PV_Gametocyte	88.11	99.86	99.34
Average of 400 Images	84.07	99.77	99.49

**Table 7 diagnostics-13-00511-t007:** The performance comparison among ELM, OS-ELM, and random forest networks for classification between malaria and non-malaria parasites.

Task	Techniques	Number of Epochs	Number of Hidden Nodes	Number of Trees	Accuracy (%)	
					**Training**	**Validation**
Classify malaria parasite and non-malaria parasite	ELM	1	6	-	83.00	83.83
	OS-ELM	1	29	-	85.37	86.39
	Random forest	-	-	32	**95.44**	**86.89**

**Table 8 diagnostics-13-00511-t008:** The performance comparison among ELM, OS-ELM, and Random Forest network for five classification tasks.

Tasks	Techniques	Number of Epochs	Number of Hidden Nodes	Number of Trees	Accuracy (%)		
					**Training**	**Validation**	**Testing**
1	ELM	1	46	-	93.86	93.81	93.33
	OS-ELM	1	44	-	94.29	93.57	93.81
	Random forest	-	-	19	**97.80**	**94.37**	**94.29**
2	ELM	1	35	-	97.45	97.25	95.98
	OS-ELM	1	46	-	97.68	97.55	**97.06**
	Random forest	-	-	22	**99.28**	**98.82**	93.73
3	ELM	1	30	-	88.05	97.41	95.17
	OS-ELM	1	41	-	89.37	97.24	94.14
	Random forest	-	-	41	**93.79**	**98.79**	**96.38**
4	ELM	1	47	-	86.74	85.91	65.45
	OS-ELM	1	40	-	86.14	**86.36**	77.95
	Random forest	-	-	33	**98.94**	84.77	**80.00**
5	ELM	1	49	-	89.12	88.46	80.00
	OS-ELM	1	49	-	89.54	89.12	**86.86**
	Random forest	-	-	33	**96.70**	**90.78**	82.25

**Table 9 diagnostics-13-00511-t009:** Comparison of our proposed classifier performance with the current relevant studies.

Authors	Task	Classifier	Dataset	Classifier Performance
Nanoti et al. [25]	Detect three lifecycle stages (trophozoite, schizont, and gametocyte)	*k*-nearest neighbor (KNN)	300 thin blood smear images	Accuracy = 90.17% Sensitivity = 90.23%
Dave [27]	Parasite detection	Support vector machine (SVM)	87 thick blood smear images	Discrepancy of parasites count = 7.18% Sensitivity = 86.34% Specificity = 96.60%
Setianingrum et al. [29]	Detect two lifecycle stages in *P. falciparum* (trophozoite and schizont)	Support vector machine (SVM)	24 images (consisting of 12 images of the trophozoite stage and 12 images of the schizont stage)	Accuracy = 91.67%
Sifat and Islam [32]	Classification of Infected and Uninfected RBC using CNN, Detect the type and stage of the detected parasite using VGG16	Convolutional neural network (CNN), visual geometry group 16 (VGG16)	5512 images (consist of 3815 *P. falciparum*, 237 *P. malariae*, 618 *P. ovale*, and 842 *P. vivax*)	Accuracy of infected RBC = 100% Specificity of infected RBC = 95% Average accuracy of species and stage identification = 95.55% Specificity of species and stage identification = 94.75%
Nugroho et al. [33]	Species recognition and staging identification	Naïve Bayes	35 thin blood smear images	Sensitivity = 84.37%, Positive predictive value = 80.60% F1-score = 80.60%
Swastika et al. [34]	Parasite detection	ResNet-50, MicroVGGNet and BaselineNet-1	27,558 thin blood smear images (consisting of 13779 infected and 13779 uninfected images)	Accuracy of ResNet-50 with regularization = 97.12% Accuracy of MicroVGGNet with regularization = 95.64% Accuracy of BaselineNet-1 with regularization = 96.28%
Taha and Liza [35]	Classification of infected and uninfected samples	InceptionResNetV2	27,558 thin blood smear images (consisting of 13,779 infected and 13,779 uninfected images)	Accuracy = 97.10% Precision = 96.93% Recall = 97.28% F1-score = 97.10% Matthews correlation = 94.20%
Widiawati et al. [36]	Identification of parasite and non-parasite	Naïve Bayes	38 thick blood smear images	Accuracy = 82.05% Sensitivity = 84.62% Specificity = 76.92%
Maqsood et al. [37]	Parasite detection	Convolutional Neural Networks	27,558 thin blood smear images (consisting of 13,779 infected and 13,779 uninfected images)	Accuracy = 96.82%
Shal and Gupta [41]	Classification of infected and uninfected samples	Yolo version 5	27,558 thin blood smear images (consisting of 13,779 infected and 13,779 uninfected images)	Accuracy = 94.67% True positivity rate = 50.30% Precision = 95.25%
Razin et al. [42]	Parasite detection and classification of infected and uninfected samples	Convolutional Neural Network (CNN) and YOLOv5 algorithm	27,558 thin blood smear images (consisting of 13,779 infected and 13,779 uninfected images)	Accuracy of infected YOLOv5 = 95.37% Accuracy of uninfected YOLOv5 = 97.05% Accuracy of parasite detection CNN= 96.21%
Zarima et al. [43]	Classification of infected and uninfected samples	EfficientNet	27,558 thin blood smear images (consisting of 13,779 infected and 13,779 uninfected images)	Accuracy = 91%
Gummadi, Ghoosh, and Vootla [44]	Parasite detection	Convolutional neural network (CNN)	27, 558 thin blood smear images (consisting of 13,779 infected and 13,779 uninfected images)	Accuracy = 96.88% Sensitivity = 97.35% Specificity = 96.41% F1-score = 96.89% Precision = 96.44%
Our proposed technique	Parasite detection, species recognition, and staging identification	Random forest	500 thick blood smear images (consisting of both *P. falciparum* and *P. vivax*), 4035 thin blood smear images (consisting of both *P. falciparum* and *P. vivax*)	Accuracy of parasite detection (thick smear) = 86.89% Accuracy of parasite detection (thin smear) = 94.37% Accuracy of species recognition = 98.82% Accuracy of staging identification = 90.78%

## Data Availability

Not applicable.

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
