# Peer review of "Robust Image Processing Framework for Intelligent Multi-Stage Malaria Parasite Recognition of Thick and Thin Smear Images"

_diagnostics, 2023, doi:10.3390/diagnostics13030511_

Round 1

Reviewer 1 Report

The diagnostic of malaria in endemic areas remain challenging despite the roll-out of antigen-based point-of-care tests. Manual microscopic examination remains the gold standard and it automatization could be helpful for the diagnostic in rural area and do not need trained technician. This work present an interest for the control of malaria in endemic areas.

I found to work original and very interested given my interest to diagnostic of infectious diseases. However, the technology used in this diagnostic is out of my competencies. So, I cannot make good recommandation to the authors.

Author Response

Thank You.

We revised properly.

Reviewer 2 Report

Comments to the Authors :

Malaria remains a serious public health problem in many countries of the world. In spite of advances in malaria diagnosis, quality assured RTDs and microscopy remain the available diagnostic tools recommended for routine malaria diagnosis of suspected cases of malaria. Issues related to sub-microscopic infections, highly skilled personnel for preparation ,staining and examination of slide as well as  sensitivity and specificity of RDTs for diagnosis of all human malaria parasites need to be addressed. In view of the existing limitations of the existing recommended diagnostic methods, new diagnostic methods are needed . The paper is important from this point of view. The work attempted  by  the authors is therefore, appreciated.

In order to make this paper presentable and comprehendible by public health specialists as well as biomedical scientists, a major revision of the paper  is recommended. Specific attention and revision of English language is required for presenting the technical details and facts appropriately. In addition, authors are advised to make specific revisions as per following comments :

1.     Abstract :

· The abstract should contain a short introduction of the topic, objective of the study  materials and methods used, results and conclusion(including a line on the future scope of the study).  

· Since some of the words like ‘Concurrently’, ‘Consequently’, ‘Accordingly’ have not been used appropriately, it is suggested that these may be omitted.

2.     Introduction:

A crisp and precise introduction to the topic is required. The first two paragraphs are lengthy and need to be revised to a short and precise introduction of the prevailing issues . Last paragraph of introduction should clearly mention the objective and scope of this study/paper.  The Line Nos. 39,43,44, 53 to 55, 63 , 64,68,73,74 need to be corrected and revised specifically for English language as well as factual information. The following sentences need special attention :

 i)Line 44 : ‘ resulting in difficulty of health and sources of  prevention’ – may be deleted.

ii)Line 53-55 : Statements like ‘Due to the large number of malaria cases, a yearly medical check-up is important as fast treatment are the keys to control and reduce the malaria mortality and morbidity’, which are incorrect factually should be deleted .

 iii)Line 68: Malaria parasite slide examination by microscopy does not take 20-60 minutes. A trained Lab. technicians is expected to examine about 45-50 slides in a day. Moreover, slide examination and reporting is routinely done by trained/expert technicians under Programmes and not by an expert microbiologist. Microbiologists might be responsible for reporting in specific private or hospital labs. in some countries.

iv)Line 71-74 is not required or may be modified with reference to specific  circumstances where the technicians are not aptly trained

v)Lines 156-164 are not required since the manuscript is expected to be written as per the instruction to authors by the journal. Instead of sections , the authors are requested to follow the conventional headings used for publication of papers in the journal  ‘Diagnostics’.

The flow of events under the Introduction can be maintained after making the contents crisp and precise but headings (sections) can be removed.      For review of previous work done also, the headings ‘ Method of--------” can be  removed and the work can be cited without specific headings. Literature review of previous work should be organized.  A table covering all work can be followed by brief description of the work . Avoiding repetitions and a scattered approach of presenting facts is recommended.

The authors may also note that the malaria parasites are not identified on the basis of colour of stains alone but the stage specific parasite morphology which includes parasite specific components . Confusion with blood cells is therefore, not as big an issue as highlighted by the authors. Revisions may be accordingly considered for providing factual information.  

3.     Material and Methods :

 The work presented in this paper has already been done, so please change future tense into past tense. What was the duration and study period(year)?

4.     Results & Discussion :

It is highly desirable that the results and discussion are written separately, ensuring that the results section focuses on the outcome of this study and correlates well with the material and methods.

5. Discussion :

Discussion should cover the interpretation of results and refer to the previous studies conducted on similar or related lines highlighting the agreement , disagreement as well as novel findings presented by the authors in this paper. Please ensure that repetitions between literature review and discussion are avoided while citing the work of author authors. Revisions in the appropriate sections under literature review of previous work should be considered accordingly to take care of this specific point.

6Conclusion:

In addition to what has been given, authors may consider giving a lead to the future work required to make this method useful as a successful diagnostic tool.

Author Response

Thank You.

We revised properly.

Reviewer 3 Report

This paper proposes an encompassing image processing and machine learning to address the encumbrances in parasite detection in thick smear microscopy, alongside multi-stage recognition of parasite species and staging for thin smear microscopy.

1- Authors should change results into two decimal places. 

2- It is good to address that some machine learning algorithms have limitations. Therefore the authors excluded them. I suggest that the author should explain the limitations of their work.

3- Some new references should be added to improve the literature review—for example, https://doi.org/10.3390/app12115500; https://doi.org/10.1186/s12859-021-04036-4.

Author Response

Thank You.

We revised properly.

Round 2

Reviewer 2 Report

A major revision of the paper was recommended to make the paper presentable and comprehensible by public health specialists as well as biomedical scientists which have been addressed by the authors to a large extent. However, the paper continues to have some unresolved issues in the English language and grammar due to which many technical facts do not get conveyed appropriately. In addition, authors were  advised to make specific revisions under different headings which have been addressed to a great extent but some changes are still required . Some new mistakes have been introduced in the revised text . Since it is beyond the capacity of the reviewer to correct every sentence, a few suggestions and corrections are suggested below for consideration of the authors:

Review/Comments (version 1)

Review/Comments (Version 2)

Abstract:

·       The abstract should contain a short introduction of the topic, the objective of the study,  materials and methods used, results and conclusion (including a line on the future scope of the study).

·       Since some of the words like ‘Concurrently’, ‘Consequently’, and ‘Accordingly’ have not been used appropriately, it is suggested that these may be omitted.

·       Not revised as suggested .

·       Corrections done and some wrong words introduced in the paper afresh , e.g.,

Line 21 : The word ‘Consequently’ has been replaced by ‘However’, which is again not correct. Please change it to ‘Therefore’.

Line 23: Please check the word  ‘ encumbrances’.

Line 31 : Pl check the word ‘Produced-parasite detection’.

Line 32 : Change ‘Accordingly’ to ‘Simultaneously’

Line 33 : Check the word parasites ’. Consider changing parasite.

Introduction:

·       A crisp and precise introduction to the topic is required. The first two paragraphs are lengthy and need to be revised to a short and precise introduction of the prevailing issues . The last paragraph of introduction should clearly mention the objective and scope of this study/paper. Lines. 39,43,44, 53 to 55, 63 , 64,68,73,74 need to be corrected and revised specifically for English language as well as factual information. The following sentences need special attention :

i)Line 44 : ‘ resulting in difficulty of health and sources of  prevention’ – may be deleted.

ii) Line 53-55 : Statements like ‘Due to a large number of malaria cases, a yearly medical check-up is important as fast treatment are the keys to control and reduce the malaria mortality and morbidity’, which are incorrect factually should be deleted .

iii)Line 68: Malaria parasite slide examination by microscopy does not take 20-60 minutes. A trained Lab. technicians is expected to examine about 45-50 slides in a day. Moreover, slide examination and reporting are routinely done by trained/expert technicians under Programmes and not by an expert microbiologist. Microbiologists might be responsible for reporting in specific private or hospital labs.in some countries.

iv)Line 71-74 is not required or may be modified with reference to specific  circumstances where the technicians are not aptly trained.

iv)Lines 156-164 are not required since the manuscript is expected to be written as per the instruction to authors by the journal. Instead of sections , the authors are requested to follow the conventional headings used for the publication of papers in the journal  ‘Diagnostics’.

·       The flow of events under the Introduction can be maintained after making the contents crisp and precise but headings (sections) can be removed.      For review of previous work done also, the headings ‘ Method of--------” can be  removed and the work can be cited without specific headings. Literature review of previous work should be organized.  A table covering all work can be followed by brief description of the work . Avoiding repetitions and a scattered approach to presenting facts is recommended.

·       The authors may also note that the malaria parasites are not identified on the basis of colour of stains alone but the stage-specific parasite morphology which includes parasite-specific components . Confusion with blood cells is therefore, not as big an issue as highlighted by the authors. Revisions may be accordingly considered for providing factual information.

·       Though some revisions have been made, issues in  technical information as well as language remain. Malaria microscopy  in lines 53-62. The information regarding parasite detection and the count is not factual. It is well documented that both thick and thin blood smears are used for parasite detection as well as counting. The authors are advised to refer WHO SOPs on malaria microscopy(2016)

Line 54-56 : The sentence ‘Here, the analysis of thick blood smears is the reference method chosen as the first option for malaria diagnosis worldwide since it is more sensitive for parasite detection’, is superfluous and doesn’t add any value.

Line 56-62: Needs factual corrections .

The thick and thin smears are visually analyzed by microscopy. The thick smear utilizes a larger amount of blood and multiple layers of RBCs due to which it is more sensitive for parasite detection.  The thin smear being made of a single layer of uniformly spread cells is more sensitive for parasite species identification. The RBCs in the thin smear are maintained intact due to alcohol fixation before staining, thereby making it possible to study the parasite morphology as well as the changes in the host RBCs which are characteristic of specific parasite species. Both RBCs, as well as WBCs, can be utilized for parasite count, though it is easier and better to use the thin smear.

Line 144-145: A new sentence has been added which reads as ‘Hence, improve standardization on thick and thin smear inspection for malaria research and potentially field diagnosis ’. The sentence is incomplete and vague. It will be better to join it with the previous sentence.

iv & v) Changes accepted.

Author Response

Thank. We revised your comment point by point. This manuscript also submitted to proofreader. Certificate as attached.

Round 3

Reviewer 2 Report

The authors have done the  changes meaningfully and the manuscript has now improved considerably, though some editing in language / grammar would still be required. The changes done in the technical part of this paper related to information on malaria are accepted.

Since I am not qualified to comment on the AI/Machine learning aspects of this paper, I am sure that the editorial board will take care of this part by inviting comments from the concerned experts for taking a final view on the publication of this paper.